# Resolution of herpes simplex virus reactivation in vivo results in neuronal destruction

Jessica R. Doll[1¤a], Kasper Hoebe[2¤b], Richard L. Thompson[1], Nancy M. Sawtell[2]*

1 Department of Molecular Genetics, Biochemistry, and Microbiology,University of Cincinnati, Cincinnati, Ohio, United States of America, 2 Division of Infectious Diseases, Cincinnati Children's Hospital Medical Center, Cincinnati, Ohio, United States of America

¤a Current address: Division of Immunobiology, Cincinnati Children's Hospital Medical Center, Cincinnati, Ohio, United States of America
¤b Current address: Division of IBD Discovery Research, Janssen R&D, Pennsylvania, United States of America
* nancy.sawtell@cchmc.org

**Data Availability Statement:** All relevant data are within the manuscript and its Supporting Information files.

**Funding:** This work was supported primarily by NASA NNX13A047G to NMS and NIH R01

## Abstract

A fundamental question in herpes simplex virus (HSV) pathogenesis is the consequence of viral reactivation to the neuron. Evidence supporting both post-reactivation survival and demise is published. The exceedingly rare nature of this event at the neuronal level in the sensory ganglion has limited direct examination of this important question. In this study, an in-depth *in vivo* analysis of the resolution of reactivation was undertaken. Latently infected C57BL/6 mice were induced to reactivate in vivo by hyperthermic stress. Infectious virus was detected in a high percentage (60–80%) of the trigeminal ganglia from these mice at 20 hours post-reactivation stimulus, but declined by 48 hours post-stimulus (0–13%). With increasing time post-reactivation stimulus, the percentage of reactivating neurons surrounded by a cellular cuff increased, which correlated with a decrease in detectable infectious virus and number of viral protein positive neurons. Importantly, in addition to intact viral protein positive neurons, fragmented viral protein positive neurons morphologically consistent with apoptotic bodies and containing cleaved caspase-3 were detected. The frequency of this phenotype increased through time post-reactivation. These fragmented neurons were surrounded by Iba1[+] cells, consistent with phagocytic removal of dead neurons. Evidence of neuronal destruction post-reactivation prompted re-examination of the previously reported non-cytolytic role of T cells in controlling reactivation. Latently infected mice were treated with anti-CD4/CD8 antibodies prior to induced reactivation. Neither infectious virus titers nor neuronal fragmentation were altered. In contrast, when viral DNA replication was blocked during reactivation, fragmentation was not observed even though viral proteins were expressed. Our data demonstrate that at least a portion of reactivating neurons are destroyed. Although no evidence for direct T cell mediated antigen recognition in this process was apparent, inhibition of viral DNA replication blocked neuronal fragmentation. These unexpected findings raise new questions about the resolution of HSV reactivation in the host nervous system.

AI093614 to NMS. Some resources from Good Venture Foundation to NMS were also utilized. The funders had no role in study design, data collection and analysis, decision to publish, or preparation of the manuscript.

## Author summary

Herpes simplex virus (HSV) is an endemic human pathogen that establishes latency in neurons and can periodically reactivate over the lifetime of the host. Whether or not neurons survive post-reactivation is controversial and has significant implications for long-term infection. HSV reactivation events can be characterized in mice, which maintain the complexity of host-pathogen interactions, with the goal to provide insight into how the virus behaves in the nervous system. In this report, it is shown that the elimination of infectious virus following reactivation in vivo corresponded with a highly focused cellular response and destruction of the neuron containing HSV proteins. The role of T cells in this response was investigated. Previous work identified T cells as major regulators of HSV reactivation. However, neuronal destruction was still observed when T cell antigen recognition co-receptors were absent and infectious virus titers and viral spread was not different from isotype treated control ganglia. Conversely, destruction of viral protein expressing neurons was not observed when viral DNA replication was inhibited. These findings suggest that reactivation is resolved through destruction of the neuron, which appears to be independent of antigen-mediated T cell cytotoxicity, but does require viral replication.

## Introduction

Herpes simplex virus (HSV) is endemic in the human population [1]. Infection occurs at the body surface, where the virus replicates in epithelial cells and is transported to sensory ganglia through innervating sensory neuron axons. HSV establishes latency in neurons, creating a reservoir with the potential to reactivate over the lifetime of the host (reviewed in: [2]). Reactivation events allow for spread to new hosts, but have also been linked to various disease sequelae, including blindness [3], encephalitis [4, 5], and potentially Alzheimer's Disease [6–9]. There is increasing evidence for a role for HSV in neurodegenerative disease (reviewed in: [8, 10]), but a mechanism whereby long term infection contributes to neurological damage is still undefined. Understanding the outcome of a reactivation event at the neuronal level is an important step towards understanding the consequences of long term HSV infection and solidifying a role in the development of neurological disease. Furthermore, identifying the host response to reactivation in the ganglia will guide vaccine and therapeutic development to prevent and/or treat HSV infection.

A quantitative, temporal analysis of HSV reactivation in vivo at the neuronal level has not been performed and the fate of the neuron post-reactivation remains controversial. Immune cell infiltrate has been detected in HSV infected human ganglia post-mortem, independent of viral protein expression [11, 12], but the role of the immune cells in directly controlling HSV reactivation events in vivo through cytolytic or non-cytolytic mechanisms is unclear. Studies utilizing dissociated trigeminal ganglia cultures derived from latently infected mice demonstrated that HSV specific CD8+ T cells can interact with latently infected neurons in a non-cytolytic manner, which suggested that T cells function to preserve the neuron and represent a potential barrier to reactivation [13–15]. However, the fate of the neurons that do successfully express viral proteins and go on to produce infectious virus was not examined [14]. Histological examination of murine ganglia post-reactivation in vivo has identified immune infiltrate surrounding HSV protein positive neurons and suggested that reactivation results in an inflammatory host response and destruction of the viral protein positive neuron [16–21].

However, the conclusion that neurons were destroyed was inferred from an extremely small number of events and based on the detection of inflammatory infiltrate, rather than a direct marker for cell death.

The resolution of in vivo reactivation occurs within a narrow time frame, which is not recapitulated ex vivo [22, 23]. Therefore, it is possible that neurons supporting reactivation may only be destroyed in the context of a living host. The mouse ocular model of HSV infection and hyperthermic stress to induce reactivation in vivo were utilized to address the outcome of reactivation in vivo. Hyperthermic stress is a physiologically relevant method for inducing reactivation, as it mimics fever, a stimulus that is correlated with HSV reactivation in humans [17, 24]. The propensity for HSV to inhibit programmed cell death (reviewed in: [25]) raised the possibility that examination of a single time point, or only a limited number of neurons post-reactivation stimulus, could substantially reduce the probability of detecting neuronal death, if it occurred as the endstage of a reactivation event. Thus, the goals of this study were to use C57BL/6 mice to (i) perform a temporal analysis of HSV reactivation at the neuronal level and (ii) to investigate the mechanism of resolution of a reactivation event in vivo. C57BL/6 mice were selected to allow for a direct comparison to previous *ex vivo* studies which concluded that CD8[+] T cells interact with neurons to impede reactivation through non-cytolytic mechanisms [14, 15]. Our data show that neurons undergoing reactivation and infectious virus titers peaked around 20 h post-reactivation stimulus and these were almost completely resolved by 48 h post-stimulus. Fragmented viral protein positive neurons, which contained cleaved caspase-3 (indicative of apoptosis), were more frequent with increasing time post-reactivation stimulus. Viral protein positive neurons were surrounded by a hypercellular cuff, which contained primarily ionized calcium binding adaptor molecule 1 (Iba1) expressing cells, a marker expressed by macrophages and microglia. These phagocytic cells are associated with removal of apoptotic neurons [26]. Treatment with anti-CD4/CD8 depleting antibodies did not lead to viral spread within the ganglia or prevent the fragmented neuron phenotype at 48 h post-reactivation stimulus. However, blocking viral DNA replication during reactivation prevented neuronal fragmentation, despite viral protein expression. These results demonstrate that there is neuronal destruction post-reactivation and that blocking viral replication inhibits this outcome.

## Results

### Reactivation occurs at a high frequency and within a narrow temporal window post-stress

The first goal of this study was to perform a temporal analysis of HSV reactivation in vivo in C57BL/6 mice. Characterization of the frequency and time frame of in vivo reactivation in the ganglion is essential for downstream identification of viral or host factors which influence reactivation in the ganglion. Two fundamental features of reactivation include: first, the entry of latently infected cells into the viral lytic cycle and second, the resulting production of infectious virus. In this study, (i) the number and context of the cells evidencing engagement of the viral lytic cycle based on viral protein expression, and (ii) the amount of infectious virus in the trigeminal ganglia (TG) was quantified at multiple time points post-stress, generating a picture of reactivation progression in the TG of C57BL/6 mice in vivo.

The uniformity and efficiency of infection with two commonly used HSV-1 strains was assessed by (i) measuring replication kinetics in the eyes, TG, and central nervous system during acute infection and (ii) quantifying the number of latent genomes in the TG at >45 days post-infection (dpi). Mice were infected on scarified corneas with either $2x10^6$ pfu of 17syn+ (males) or $2x10^5$ PFU of McKrae (females). Replication peaked at 4 dpi for both viruses,

although McKrae replicated to higher titers than 17syn+ (S1A Fig; Student's t-test; p = 0.0897 and S1B Fig; Student's t-test p = 0.0292). 345 mice were utilized over the course of these studies and infected in groups of 30–40. Infection with McKrae resulted in mortality in 41% of female C57BL/6 mice while only 4% of male C57BL/6 mice infected with 17syn+ succumbed to infection. This difference in mortality between the HSV-1 strains McKrae and 17syn+ was significant (Log-rank [Mantel-Cox]; p<0.0001) (S1C Fig) and was reflected in viral titers in the CNS (S1D Fig). To determine the reproducibility of infection, infectious virus in tear films during acute infection and/or in eyes and TG at 4 dpi were analyzed for each infected group. The infectious titers were consistently within the ranges represented in S1 Fig, emphasizing the uniformity among the biological replicates with respect to infection and acute replication.

Although it is common practice to choose 28–30 days post-infection as "latency", testing for the establishment of latency remains critical. Low levels of infectious virus in groups of mice prior to the induced reactivation stimulus would confound the interpretation of the frequency of induced reactivation. At 30 dpi, TG were assayed for infectious virus. Infectious virus was recovered from 2/5 mice (2 and 3 pfu) infected with McKrae, which suggested that there remained a high frequency of spontaneous reactivation at this time [20]. At 45 dpi, TG from 5 mice latently infected with McKrae were again tested for the establishment of latency and at this time no infectious virus was recovered (0/5 mice). TG from mice latently infected with 17syn+ were also tested at 30 dpi and 45 dpi. Infectious virus was recovered from 0/5 and 0/4 mice, respectively. All reactivation experiments were performed at greater than 45 dpi.

The number of HSV genomes in the latently infected TG (>45 dpi) was quantified by Real Time qPCR, as described in Methods [27]. Similar levels of latent viral establishment were observed between animals infected with the same viral strain, however, mice infected with 17syn+ showed a 10-fold increase in viral genomes compared to mice infected with McKrae ($4.2 \times 10^3 \pm 1.2 \times 10^3$ vs. $4.2 \times 10^2 \pm 4.3 \times 10^1$ genomes/50ng DNA, respectively; Student's t-test; p = 0.0057).

In vivo reactivation was induced in latently infected C57BL/6 mice (>45 dpi), using hyperthermic stress [17]. Reactivation was quantified at the indicated times post-stress by measuring infectious virus titers in the TG and by counting the number of viral protein expressing neurons in each ganglia using whole ganglia immunohistochemistry. This second approach localized the reactivation events within the ganglion and provided information on the morphology of the reactivating cell. Analysis of groups of latently infected mice prior to hyperthermic stress demonstrated that neither infectious virus (0/5 mice) nor viral protein expressing cells (0/10 TG) were detectable in the TG at this time (Fig 1). At 20 h post-hyperthermic stress (phs), infectious virus was recovered from 78% of TG latently infected with strain McKrae (range 1–149 pfu) (Fig 1A) and 62% of TG latently infected with strain 17syn+ (range 2–23 pfu) (Fig 1B). HSV proteins were detected in neurons in 70% of TG latently infected with strain McKrae (range 1–12 neurons) and 77% of TG latently infected with strain 17syn+ (range 1–10 neurons/ganglion), at 20 h phs (Fig 1C and 1D). The frequency at which infectious virus was detected was very similar to the frequency of ganglia containing viral protein positive neurons at each time point. After 20 h phs, infectious virus titers and the number of neurons positive for viral protein declined with increasing time post-hyperthermic stress (Fig 1A–1D). Consistent with previous reports [17, 23], reactivation was largely resolved by 48 h phs (Fig 1A–1D). This narrow window in which production and elimination of infectious virus occurs provides a time frame for visualizing the impact of viral reactivation at the neuronal level and analysis of the resolution of reactivation.

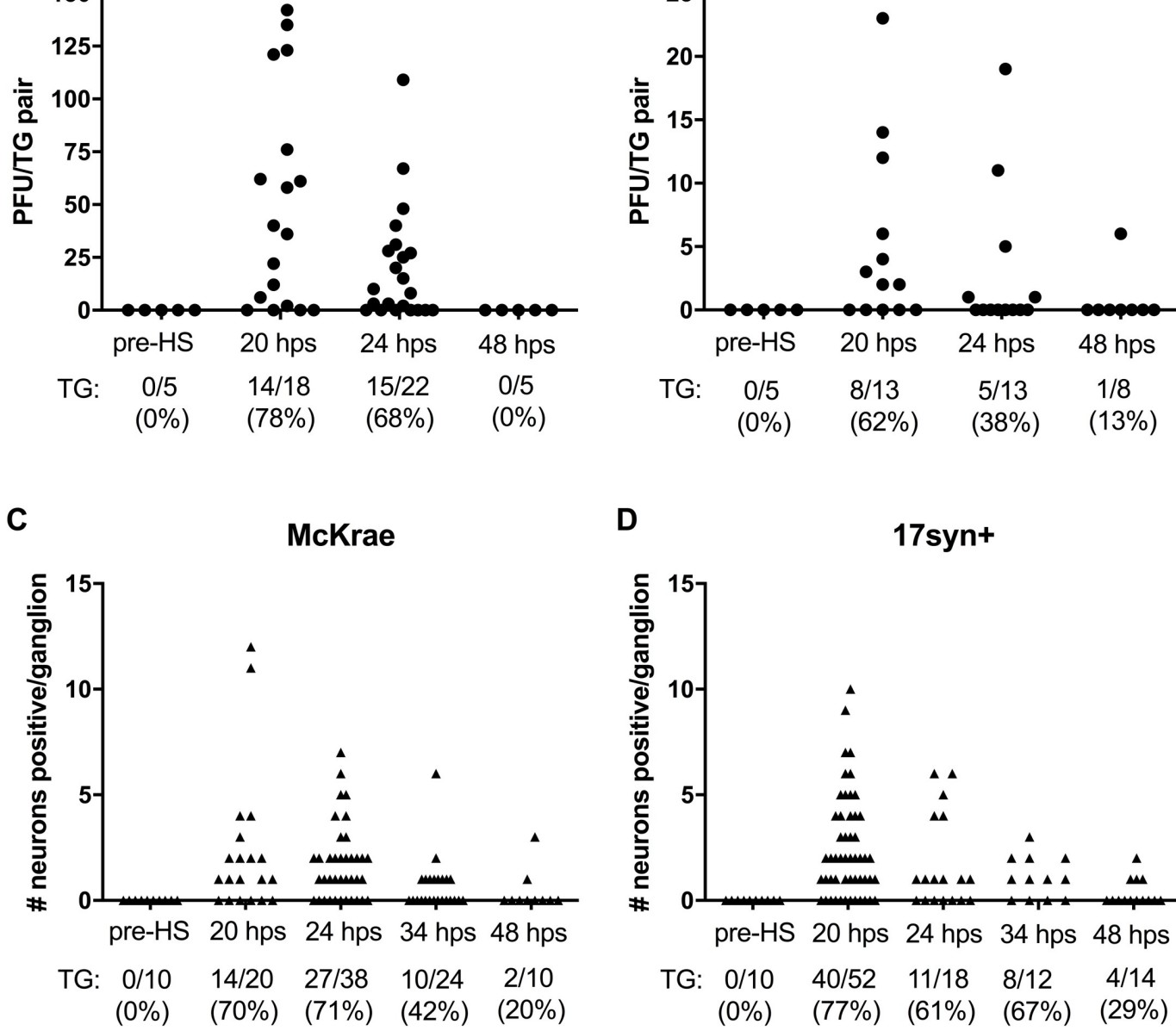

**Fig 1. In vivo reactivation in C57BL/6 mice.** Latently infected mice were subjected to hyperthermic stress and trigeminal ganglia were harvested at the indicated times. (A and B) The amount of infectious virus recovered in TG pairs pre- and post-hyperthermic stress from C57BL/6 mice latently infected with (A) McKrae or (B) 17syn+. Each point represents the total amount of infectious virus recovered per TG pair. The percent reactivation (number of TG positive/total TG evaluated) is given below each graph. (C and D) TG were processed for viral protein expression in whole ganglia pre- and post-hyperthermic stress as described in Methods from mice latently infected with (C) McKrae or (D) 17syn+. Each point represents the total number of neurons positive for HSV protein expression in a single ganglion. The percent exit from latency (number of TG positive/total TG evaluated) is given below each graph.

## Resolution of reactivation correlates with apoptosis of HSV positive neurons

The second goal of this study was to investigate the mechanism of resolution of a reactivation event. Immunohistochemical analysis of whole ganglia revealed that in addition to the

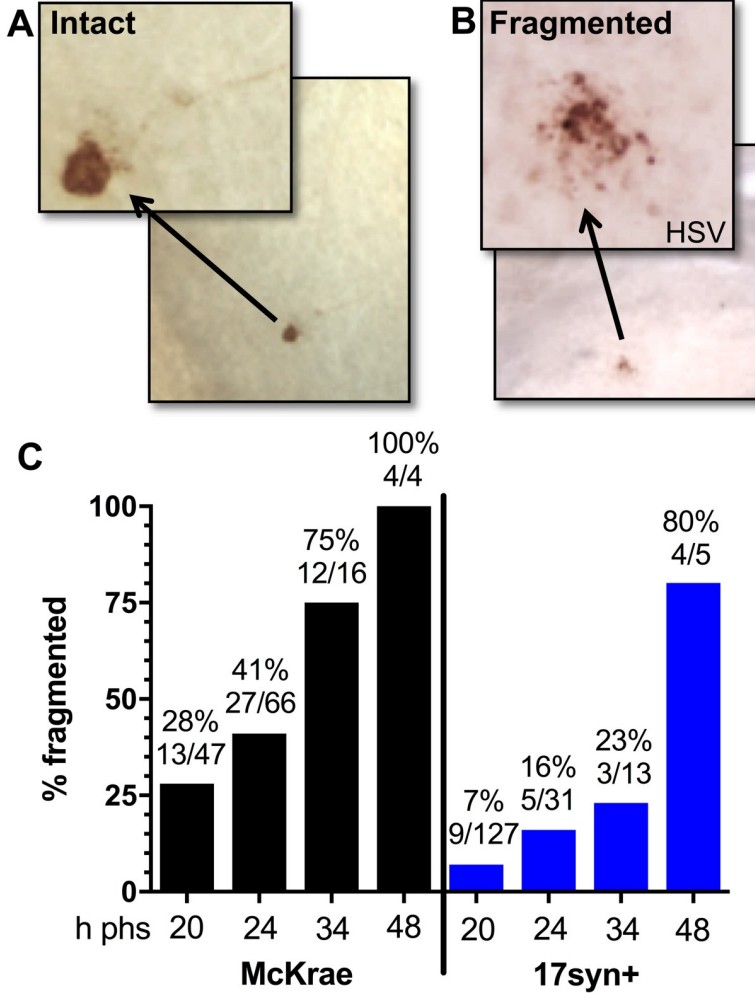

**Fig 2. Neuronal fragmentation over time post-reactivation in vivo.** Latently infected mice were subjected to hyperthermic stress and ganglia were harvested at the indicated times post-hyperthermic stress (minimum of 10 ganglia per time point, see Fig 1C and 1D). Whole ganglia were processed for HSV protein expression. The brown precipitate (DAB) marks viral exit from latency. (A and B) Photomicrographs of portions of whole mount TG showing viral protein positive neurons that are (A) intact or (B) fragmented. Arrows depict higher magnification of neurons. (C) Neurons represented in Fig 1C and Fig 1D were evaluated with respect to the intact or fragmented phenotype. The percentage of fragmented neurons out of the total number of antigen positive neurons is indicated at each time post-hyperthermic stress examined. Fragmentation increased significantly over time post-reactivation in both McKrae (Chi square; p = 0.0008) and 17syn+ (Chi square; p<0.0001) infected TG.

morphologically intact viral protein positive neurons (Fig 2A), fragmented viral protein positive neurons (Fig 2B) were observed in both McKrae and 17syn+ latently infected TG. Two alternatives are possible. First, that the two phenotypes represent separate populations, suggesting that (i) the expression of viral protein and the fragmentation occur concurrently and (ii) the intact neurons expressing viral protein abruptly disappear. The second alternative is that the set of fragmented neurons arise from the intact set of neurons and that the fragmentation and death of the neuron is the end stage of a reactivation event. If the fragmented phenotype represented the end stage of an individual reactivation event, the percentage of fragmented reactivating neurons would inversely correlate with infectious viral titers in the ganglia. Therefore, the percentage of fragmented HSV positive neurons was determined at each time point post-stress. The proportion of the viral protein positive neuron population

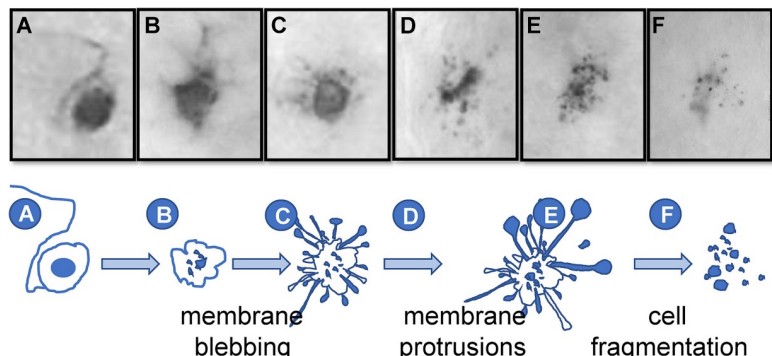

**Fig 3. Proposed progression of neuronal destruction during reactivation in vivo.** Latently infected mice were subjected to hyperthermic stress and ganglia were harvested 20–48 h phs. Whole ganglia were processed for HSV protein expression. The dark precipitate (DAB) marks viral protein expression. (A-F) Based on the well-characterized progression of cells undergoing apoptosis, a schematic of the proposed progression from an intact to fragmented state of TG neurons undergoing HSV reactivation is shown. Photomicrographs of actual viral protein positive neurons that are: (A) intact, (B) early membrane blebbing, (C) prominent membrane protrusions, (D) fully fragmented with prominent apoptotic bodies, and (E and F) fully fragmented with a small cluster of apoptotic bodies remaining are shown above the schematic and correspond with the letters indicated at each stage on the schematic.

that was fragmented increased from 28% at 20 h phs, to 41% at 24 h phs, to 75% at 34 h phs, and to 100% at 48 h phs in TG latently infected with strain McKrae (Chi-square; p = 0.0008) (Fig 2C). Fragmentation also increased from 7% at 20 h phs, to 16% at 24 h phs, to 23% at 34 h phs, and to 80% at 48 h phs in TG latently infected with strain 17syn+ (Chi-square; p<0.0001) (Fig 2C). In addition, there is an inverse relationship between the two neuronal phenotypes (S2 Fig). The proportion of intact neurons decreased while fragmented neurons increased between 20–48 hours post-stress, in support of the transition hypothesis. The presence of intermediate phenotypes between intact and fragmented further suggested that intact neurons transition to the fragmented phenotype (Fig 3). The total number of intact HSV protein positive neurons in the group over the time examined was 77 and the number of fragmented only 56. This could indicate that not all intact neurons proceed to the fragmented phenotype, however discordant lifespans of the phenotypes could also underlie this difference. Once the neuron transitions into the fragmented phenotype, relatively rapid clearance of the apoptotic bodies would be expected, reducing the probability that this phenotype would be detected, compared to the longer lived intact phenotype.

This fragmented phenotype was not restricted to either the C57BL/6 mouse strain or hyperthermic stress. Fragmented neurons were observed in two additional variations of the ocular model including (i) hyperthermic stress induced reactivation in male Swiss Webster mice latently infected with strain 17syn+ and (ii) scarification (local physical trauma) induced reactivation [28] in female Swiss Webster mice (S3 Fig). These findings are consistent with the hypothesis that the fragmented phenotype represents a common end stage of the reactivation process at the neuronal level. Based on the examination of neurons undergoing reactivation in TG post-stress, a likely progression from intact to fragmented neuronal phenotype is presented in Fig 3. These fragmented neurons were morphologically consistent with apoptotic bodies [29], which are membrane bound vesicles of cellular debris resulting from apoptotic cell death [30]. Whether this phenotype represented a late stage of apoptosis was investigated further.

HSV positive neurons post-reactivation are extremely rare events and while the percentage of viral protein positive neurons that were fragmented was high at 48 h phs (80–100%), the absolute number was quite low (9 neurons/ 24 TG) (Fig 1C and 1D, Fig 2). As determined from the temporal analysis (Fig 1C and 1D), 34 h phs represented a late time when the

frequency of detection of reactivation was declining (42–67%), but compared to 48 h phs, more HSV positive profiles remained. To determine whether the fragmented neurons were undergoing apoptosis, expression of the apoptotic executioner enzyme cleaved caspase-3 (reviewed in: [29]) was examined in TG harvested at 34 h phs from mice latently infected with strain McKrae (Experiment 1: n = 6; Experiment 2: n = 5). For this analysis, 10 μm serial sections were assayed for expression of viral proteins or cleaved caspase-3 (Fig 4A, 4B, 4D and 4H). The thicker sections facilitated visualization of the fragmented neurons, which were more easily seen as aggregated bodies in the unsectioned tissue (whole ganglia IHC, Fig 2B), but in thin sections, were reduced to unremarkable, highly focal punctate staining (Fig 4B and 4C). Fig 4 is constructed to show the localization of viral proteins and cleaved caspase-3 in the same area on serial sections. Both high and low power views are provided so that both detailed and an overall distribution of staining within the ganglia is revealed. Panels G and H are low power views of panels B and F. Regional landmarks are marked with '*' or 'X' to facilitate orientation. These analyses revealed an absence of cleaved caspase-3 in intact neurons positive for viral protein (0/5) (Fig 4A and Inset), while all viral protein positive fragmented neurons (Fig 4B) were also positive for cleaved caspase-3 (3/3) (Fig 4D).

Adult neurons are resistant to apoptosis, however some insults do result in neuronal death by apoptosis or other death pathways [29, 31, 32]. Clearance of cellular debris in the peripheral ganglia has been shown to be mediated by macrophages and resident glial cells, such as satellite glial cells, Schwann cells, and microglia (reviewed in: [33]). Microglia can be classified based on morphology [34, 35], which is closely related to functional state [36]. Activated, phagocytic microglia and macrophages, that would be anticipated to respond to the end stage of neuronal destruction, express Iba1 [26]. We asked whether Iba1 expressing cells were spatially related to the neurons undergoing reactivation. While small clusters of Iba1+ cells were detected in association with neurons expressing viral proteins, extensive layers of these cells hallmarked the region around fragmented neurons (Fig 4E and 4F). These observations are consistent with HSV reactivation being resolved through apoptosis late in the reactivation process.

## General localization of Iba1+ cells in ganglia of mock and HSV latent infection

In order to put the post-reactivation distribution of Iba1+ cells in context, the presence and localization of Iba1+ cells was determined on sections from both uninfected ganglia and mock-infected ganglia, obtained pre- and at 24 and 48 h phs, and latently infected ganglia pre- and at 34 h phs. Thin, elongated Iba1+ cells were distributed along the axonal tracts as well as around neuronal bodies (Fig 5A). The morphology and distribution of Iba1+ cells in uninfected and mock-infected TG both pre- and post-hyperthermic stress was not different. In contrast, in latently infected ganglia, there were regional variations in the pattern and morphology of the Iba1+ cells. Although Iba1+ cell morphology resembled uninfected control ganglia in some areas, in others, an increased density of rounded Iba1+ cells was observed (Fig 5B). In areas primarily along the tracts, enlarged vacuolated (foamy) cells were widely distributed (Fig 5C). These altered morphologies are indicative of activated microglia [37]. The Iba1+ cells were densely clustered together as nodules that filled spaces where neurons would be anticipated to have been located (Fig 5D).

The regional variation of Iba1+ cells in latently infected ganglia prompted examination of Iba1+ cell localization at the whole ganglia level. Uninfected control ganglia and additional latently infected ganglia, collected at 34 h phs, were processed by whole ganglia IHC for Iba1 expression. Evenly distributed Iba1+ cells among neuronal bodies and along axonal tracts were observed in uninfected whole ganglia, as observed in the uninfected sectioned ganglia.

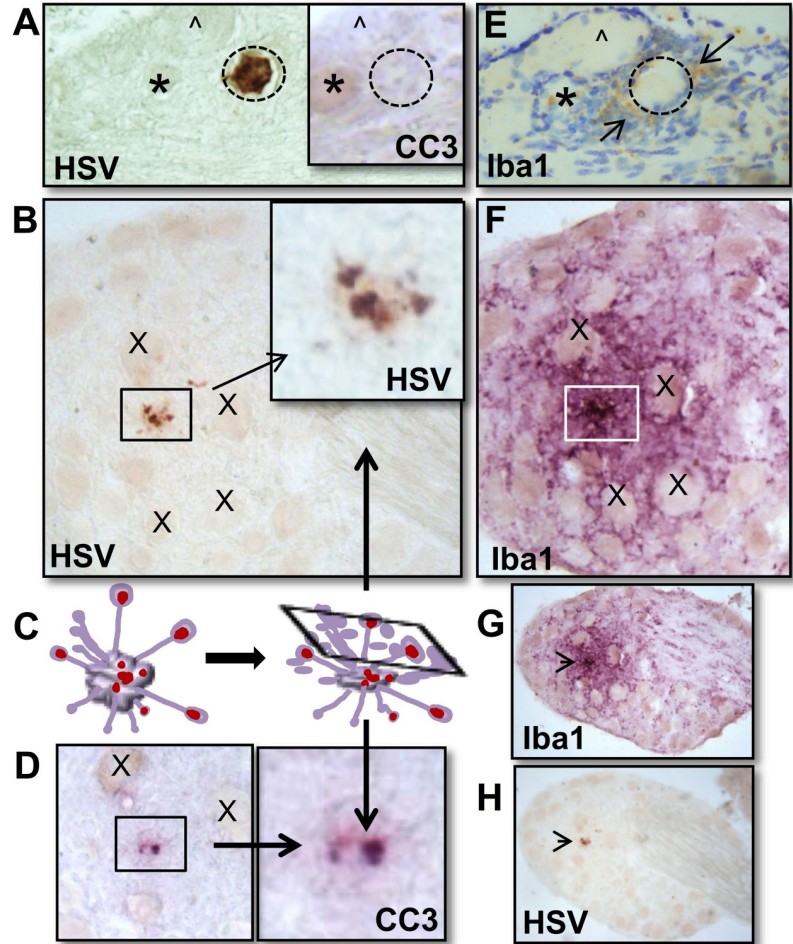

**Fig 4. Colocalization of cleaved caspase-3 with fragmented HSV+ neurons, but not intact HSV+ neurons.** Latently infected mice (n = 6; n = 5) were subjected to hyperthermic stress and ganglia were harvested 34 h phs. All ganglia were directly fixed in 4% formaldehyde, embedded, sectioned, and serial sections were assayed for HSV, cleaved caspase-3, or Iba1 expression as described in Methods. (A) Intact HSV positive (DAB; brown) neuron detected in sectioned material did not express cleaved caspase-3 (cc3). A serial section revealed few Iba1+ (brown) cells surrounding the intact neuron undergoing reactivation (cresyl violet counterstain). (*) and (^) mark the same neurons in each section to provide orientation. (B) Fragmented HSV positive (brown) neuron shown at a higher magnification in the box. This neuron was surrounded by extensive Iba1+ microglia (VIP; purple). (X) marks nearby neurons to provide orientation. A lower magnification view of both sections is provided with an arrow indicating the HSV positive fragmented neuron. (C) Pictorial representation of a neuron undergoing apoptosis and characteristic membrane protrusions. The rectangle represents a cross section through this cell that corresponds with the observed cellular fragments in (B and D). (D) A serial section of the fragmented HSV positive neuron in 'B' was found to express cc3 (VIP; purple). A higher magnification of the boxed area is shown. (X) marks the same neurons shown in 'B'. (E) A serial section of the intact HSV positive neuron in 'A' was found to have some Iba1+ cells (brown) present in the surrounding inflammatory cuff following counterstaining with cresyl violet. (F) A serial section of the fragmented HSV positive neuron in 'B' was found to have many Iba1+ cells (VIP; purple) present in the surrounding inflammatory cuff. (G) A lower power view of 'F' and (H) a lower power view of 'B' demonstrate the localized specificity of reactivation and the formation of the inflammatory cuff in the ganglia.

However, in latently infected ganglia at 34 h phs, localized clusters of Iba1+ cells were observed at variable densities in regions containing neuronal bodies (Fig 5E). This approach also revealed clusters of Iba1+ cells at intervals along an axonal tract (Fig 5F), suggestive of a response to reactivated virus potentially egressing at varicosities during transport to the body surface [38, 39].

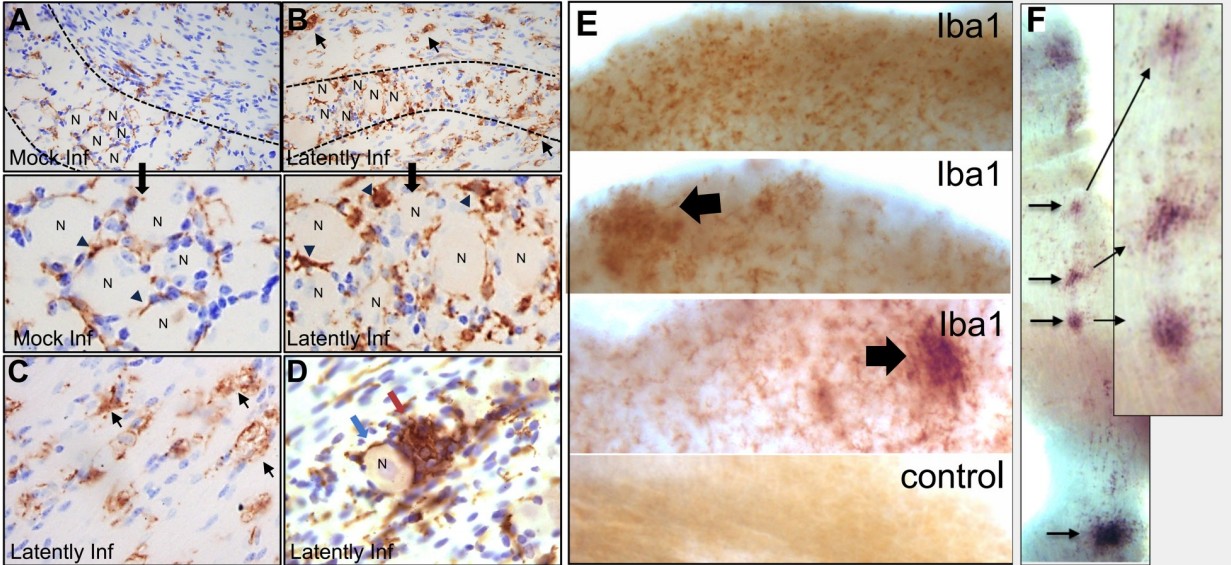

**Fig 5. Detection of Iba1⁺ microglia in HSV infected and mock infected ganglia.** Ganglia (A-D) were directly fixed in 4% formaldehyde, embedded, sectioned, and assayed for Iba1 expression as described in Methods. (A and B) Images of representative sections through a mock-infected (A) or latently infected (B) ganglion. Dashed lines indicate the border between the axonal and neuronal regions of the ganglion. Thin elongated cells expressing Iba1 (brown DAB reaction product) are present around neuronal bodies and along the axonal tracts (arrowheads) throughout the mock infected ganglion (A). In the latently infected ganglion, Iba1 expressing cells exhibit distinct morphologies, including large, round cells and vacuolated (foamy) cells along the axonal tracts (B and C) (arrows). "N" indicates a neuronal cell body. Clusters of neurons are marked in both A and B and shown at higher magnification. (D) Nodule of Iba1⁺ cells (brown) in latently infected ganglia pre-hyperthermic stress (red arrow) juxtaposed to a morphologically intact neuron (blue arrow). (E-F) At 34 h phs, ganglia were fixed in 0.5% formaldehyde and assayed for Iba1 expression in the whole ganglia, as described in Methods. (E) Low power view showing examples of cells expressing Iba1 protein (brown staining) and a control that was processed without primary antibody. Arrows indicate regions of dense clusters of Iba1 expressing cells found to be associated with viral reactivation (see Fig 6). These dense foci were not present in every ganglion and their numbers paralleled the number of reactivating neurons. (F) Clusters of Iba1⁺ cells (VIP; purple) were observed lining an axonal tract. Arrows point to the localization of Iba1 clusters at higher magnification.

## Infiltrate surrounding viral protein positive neurons represents a dynamic, time-dependent process potentially related to neuronal destruction

It has previously been shown that in vivo, a heterogenous population of immune cells surround viral protein positive neurons post-reactivation [16–19]. However, the number of reactivating neurons examined in these studies was limited and a quantitative, temporal analysis of inflammatory cell association with HSV protein positive neurons post-stimulus has not been described. To assess whether the neurons we detected undergoing reactivation were uniformly associated with a localized inflammatory response and whether this response was temporally regulated, ganglia that had been processed by whole ganglia IHC were subsequently embedded in paraffin and sectioned at 10 μm. All serial sections were systematically scanned to identify rare viral protein positive neurons (Fig 6A). Regional landmarks were used to identify fragmented neurons in these sections that were first observed in the whole ganglia. A total of 78 HSV protein positive neurons detected in whole ganglia at the indicated times post-reactivation were captured on sections and evaluated using cresyl violet staining to add surrounding cellular context (Table 1). HSV protein positive neurons were considered to be associated with a hypercellular cuff if a zone of four or more cells deep surrounded the neuron (Fig 6B).

The number of viral protein positive neurons associated with a hypercellular cuff increased over time post-stress (Table 1). At earlier times (20–24 h phs), a subset of intact neurons lacked a cuff (Fig 6A), however by 48 h phs all viral protein positive neurons were surrounded by an

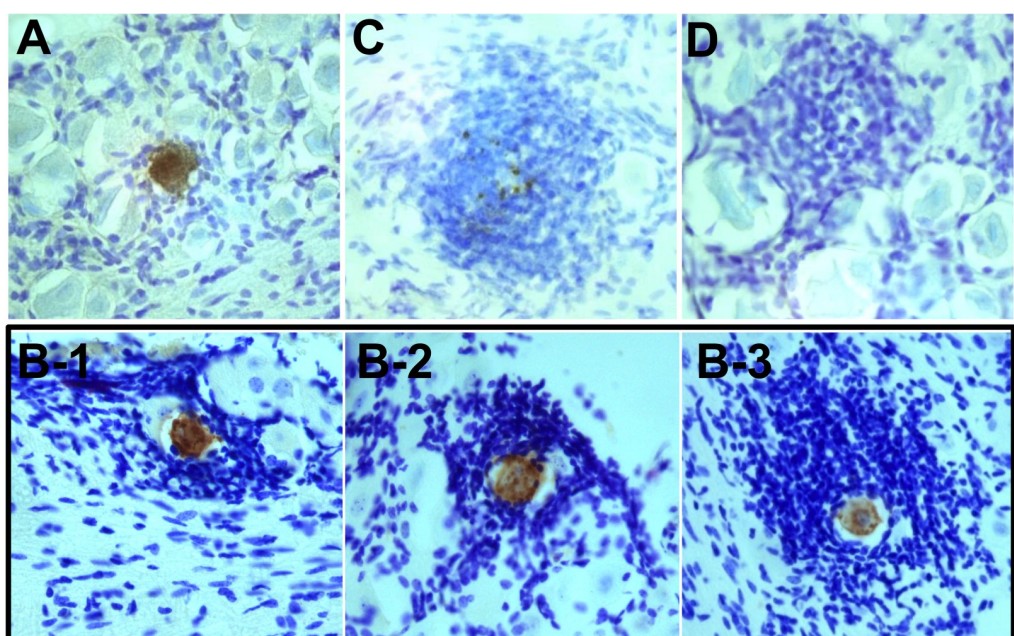

**Fig 6. HSV reactivation is associated with an inflammatory response and neuronal destruction.** Latently infected mice were subjected to hyperthermic stress and ganglia were harvested at 20–48 h phs. HSV proteins were first detected in the whole ganglia and TG were subsequently embedded, sectioned at 10 μm, and counterstained with cresyl violet, as described in Methods. (A) HSV protein positive neuron (brown) not surrounded by inflammatory infiltrate. (B) 3 examples of intact HSV antigen positive neurons (brown) surrounded by various sizes of inflammatory cuffs in the ganglia following reactivation in vivo. (C) Fragmented HSV protein positive neuron (brown) surrounded by an inflammatory cuff. (D) Nodule of cells in a neuronal region of the ganglia in the absence of viral protein detection.

inflammatory cuff (6/6 profiles) (Fig 6B) (Table 1). Regardless of time post-stress, all of the neurons with a fragmented phenotype were surrounded by a hypercellular cuff (12/12 neurons) (Fig 6C) (Table 1). HSV strain 17syn+ latently infected TG, compared to strain McKrae, showed a delay in timing of cuff formation (p = 0.039; Fisher's exact test) (Table 1). The fact that the presence and size of cellular cuffs was variable at 20 h phs suggested that recruitment was not a synchronous process related to the absolute time post-stress. As observed with Iba1$^+$ localization in the latently infected ganglia post-hyperthermic stress, hypercellular nodules

**Table 1. Quantification of HSV protein positive neurons and association with an inflammatory cuff post-hyperthermic stress of C57BL/6 mice latently infected with McKrae or 17syn+.** Mice latently infected with McKrae or 17syn+ were subjected to hyperthermic stress and ganglia were harvested at the indicated times post-hyperthermic stress (phs) (n = 5–7 per experiment; results pooled from 2 independent experiments at each time point). HSV proteins were first detected in the whole ganglia and TG were subsequently embedded, cut as 10 μm sections, and counterstained with cresyl violet, as described in Methods. Each HSV protein positive neuron in the sectioned material was then related back to the whole ganglia tissue analysis to confirm the identity as "intact" or "fragmented" and assessed for association or lack of association with an inflammatory cuff, determined as ≥4 inflammatory cell radius (see Fig 6).

| McKrae, 17syn+ (h phs) | | 20 | 34 | 48 |
|---|---|---|---|---|
| **Intact** | Cuff | 28*,10** | 1*,6** | 0*,1** |
| | No cuff | 8*,11** | 0*,1** | 0*,0** |
| **Fragmented** | Cuff | 3*,0** | 2*,2** | 2*,3** |
| | No cuff | 0*,0** | 0*,0** | 0*,0** |

\* numbers observed in McKrae infected ganglia

\*\* numbers observed in 17syn+ infected ganglia

were detected in spaces that would have been anticipated to previously contain neurons (Fig 6D) [40]. More refined analyses regarding the dynamics of inflammatory cuff composition will require the ability to track individual neurons in vivo over time. Despite this limitation, we can conclude that at increasing times post-stress there was an increase in the recruitment of cells surrounding neurons undergoing reactivation.

## Depletion of CD4 and CD8 T cells

The above results emphasize the highly restricted nature of HSV reactivation at the neuronal level and the absolute control of spread of virus within the ganglion. This is in stark contrast to explanted latently infected ganglia where virus spreads freely [22, 23]. The role of T cells in viral infection in general, and in the control of HSV infection is well established [41–43]. During acute infection, in vivo depletion of CD4 and CD8 T cells resulted in dramatically increased HSV infectious viral titers and spread [44]. A role for CD8+ T cells, and indirectly CD4+ T cells in controlling reactivation has come from several lines of evidence [14, 15]. However, whether CD4/CD8 T cells directly modify the exit from latency or outcome of the neuron undergoing reactivation in vivo remains an open question. We addressed this question by asking whether CD4 and CD8 depletion altered the number of neurons undergoing reactivation, the spread of virus within the ganglion, or fragmentation of viral protein positive neurons.

The widely utilized depleting anti-CD4 and -CD8 antibodies had a dramatic effect on viral replication following a single dose 1 dpi. Compared to mice treated with isotype control antibody, at 7 dpi 41-fold more infectious virus was recovered in the eyes (Student's t-test; p = 0.019) and 624-fold more infectious virus was recovered in the TG (Student's t-test; p = 0.025) of anti-CD4/CD8 treated (S4A Fig). A comparison of HSV protein expression in the TG between groups at this time paralleled infectious virus titers. Large numbers of viral protein expressing neurons in the anti-CD4/CD8 group contrasted with the few remaining in the control group (S4B and S4C Fig). Viral proteins detected along axonal tracts suggested that active replication and transport of virus between the TG and the epithelial surface was ongoing (S4C Fig).

During acute infection, circulating T cells enter the TG and establish a tissue resident population [45, 46]. Latently infected mice therefore have both central and tissue resident T cell populations with the potential to influence the reactivation outcome. Thus the effect of anti-CD4/CD8 treatment on T cells in the TG and spleen was examined. Near complete loss of CD4 and CD8 markers was observed in the spleen and TG when mice were treated with anti-CD4/CD8 antibodies at 3 days and again at 1 hour prior to hyperthermic stress (evaluated at 48 h phs) (Fig 7A). In isotype treated control mice, the majority of the CD3, TCRβ double positive population was either CD4 or CD8 positive. In anti-CD4/CD8 treated mice, an increase in CD4, CD8 double negative cells that were CD3, TCRβ positive was detected (Fig 7B). This suggested that anti-CD4/CD8 treatment depleted many, but not all, T cells in the ganglia and in addition resulted in loss of surface CD4 and CD8 coreceptors in those cells not depleted. These cells were examined for surface binding of rat IgG which would indicate that depleting antibodies were not simply blocking the detection of the coreceptors.

To test whether loss of the CD4/CD8 receptors correlated with loss of T cell function, in vitro and in vivo analyses were performed. T cells recovered from infected mice were tested by ex vivo antigen stimulation in the presence or absence of depleting or isotype matched control antibodies, as detailed in Methods. In this assay, anti-CD4/CD8 reduced T cell production of IFNγ in response to HSV antigen, confirming that treatment with these antibodies inhibits specific antigen stimulated T cell function in this setting (S5 Fig). Dendritic cell cross presentation of antigen from inactivated virus to T cells is not highly efficient, so to evaluate the in vivo

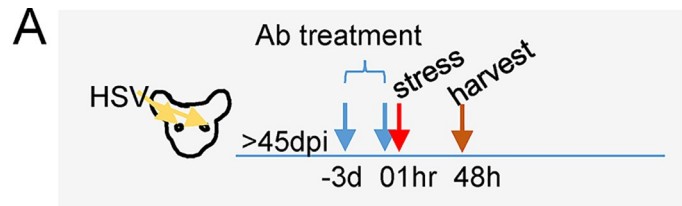

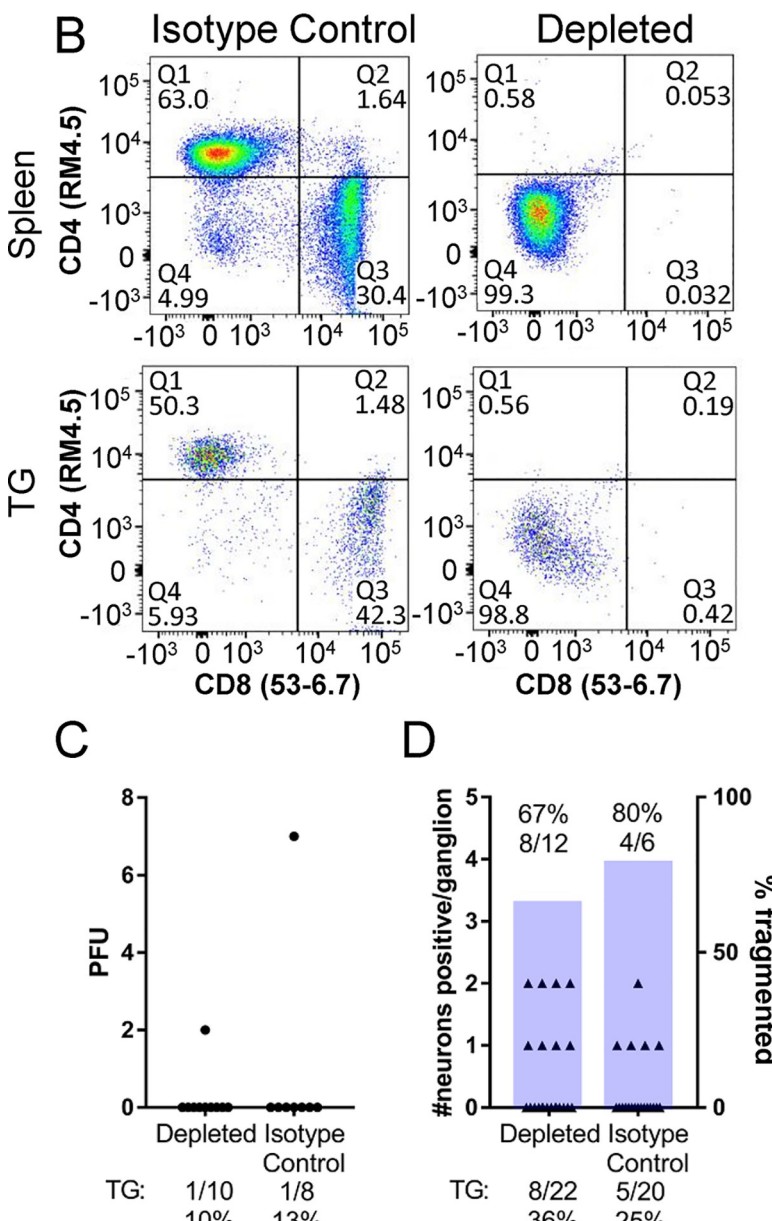

**Fig 7. T cell depletion prior to induced reactivation.** Mice latently infected with HSV-1 17syn+ were treated with depleting antibodies 3 days prior to hyperthermic stress and given a second dose 1 h prior to hyperthermic stress, as described in Methods. Tissues were harvested at 48 h phs. (A) Schematic representation of experimental approach. (B) Depletion of CD4 and CD8 was confirmed by flow cytometry of the spleen and TG. (C) Infectious virus recovered in TG from depleted and control mice (Student's t-test; p = 0.41). The percent reactivation (number of TG positive/total TG evaluated) is given below the graph. (D) The number of HSV positive neurons per ganglion in depleted and control mice are plotted on the left y-axis (Fisher's Exact test; p = 0.51) and the percent exit from latency (number of TG

positive/total TG evaluated) is given below the graph. The number of fragmented neurons out of total HSV positive neurons are represented as a bar corresponding to the right y-axis (Fisher's Exact test; p> 0.99) and the absolute number is given above each bar.

functional effect of anti-CD4/CD8 treatment during latency, an intracellular cytokine detection assay using brefeldin A (BFA) to prevent cytokine release was utilized [47]. This allowed us to test whether IFNγ production was inhibited by anti-CD4/CD8 treatment in vivo. In this assay, re-exposure to HSV by infection of latently infected mice rather than induced reactivation was utilized to ensure viral antigen exposure in all mice. Since the anti-CD4/CD8 treatment was the same, we reasoned that this approach would provide a relevant functional readout of the anti-CD4/CD8 treatment. Latently infected mice were treated with either anti-CD4/CD8 antibodies or isotype control antibodies at 3 days and again at 1 hour prior to HSV re-infection. BFA was injected 24 h later and after 6 h, tissues were removed and the intracellular accumulation of IFNγ was determined as detailed in Methods. Naïve mice treated with these same antibodies were examined in parallel as a reference for background levels of IFNγ detection. IFNγ production was detected in spleens and TG of latently infected isotype control mice re-exposed to HSV, however IFNγ production was not detected in mice treated with anti-CD4/CD8 antibodies (S6 Fig), providing further evidence that antigen mediated T cell function was inhibited by the depleting antibodies administered.

To test the role of CD4 and CD8 T cells in the resolution of in vivo HSV reactivation, mice latently infected with 17syn+ were treated with anti-CD4/CD8 antibodies 3 days prior to and the day of the reactivation trigger, as described in Methods. If antigen dependent cytotoxic T cell function was central in the immediate control of HSV reactivation events, an increase in infectious virus production and spread of infection to neighboring cells, as opposed to neuronal fragmentation and containment of the reactivation event, would be anticipated. Results pooled from 2 independent experiments are shown in Fig 7C and 7D. Surprisingly, no differences in these parameters in ganglia from anti-CD4/CD8 treated mice compared to control ganglia at 48 h phs were observed. Infectious virus was not recovered from the eyes of isotype control treated mice (0/8) and treatment with anti-CD4/CD8 antibodies did not alter this outcome (0/10). Infectious virus was recovered from the TG of 1/8 (7 pfu) and 1/10 (2 pfu) of isotype treated control and anti-CD4/CD8 treated mice, respectively (Student's t-test; p = 0.41) (Fig 7C). Viral proteins were detected in 5/20 and 8/22 control and anti-CD4/CD8 treated TG at 48 h phs, respectively (Fisher's Exact test; p = 0.51) (Fig 7D). The number of fragmented neurons was not different between control and anti-CD4/CD8 treated TG, 4/6 and 8/12 neurons displayed this phenotype, respectively (Fisher's Exact test; p> 0.99) (Fig 7D). Anti-CD4/CD8 treatment did not influence reactivation in vivo, either by increasing numbers of neurons exiting latency, the amount of infectious virus produced, or disrupting viral containment within the neuron resulting in cell to cell spread, as observed during acute infection (S4C Fig). In a separate experiment, extended treatment with anti-CD4/CD8 antibodies by dosing mice every 5 days over the course of a month did not alter reactivation outcome; HSV proteins were detected in only 1 neuron (which was fragmented) in 1/12 TG at 48 h phs, similar to ganglia from control treated mice.

These findings suggested that CD4 and CD8 T cells play a limited or redundant role in regulating the direct outcome of a HSV reactivation event. In order to explore this further, we asked whether abortive reactivation would trigger fragmentation in the same time frame as complete reactivation. We have shown previously that treatment with acyclovir (ACV) to inhibit viral DNA replication during reactivation in vivo prevents infectious virus production but does not prevent viral protein expression [27, 48]. This approach would reveal whether

fragmentation is triggered by viral protein expression per se (for example through antigen presentation and immune signaling), or requires downstream events such as changes in neuronal homeostasis linked to viral DNA replication. The major immunodominant determinant following HSV infection, gB [49], is expressed early and independent of DNA replication [50]. Therefore expression of gB and stimulation of CD8+ T cells would be anticipated in the context of ACV treatment during HSV reactivation.

Latently infected mice were treated with acyclovir (ACV; 50mg/kg delivered via intraperitoneal injection), or vehicle alone, starting 24 h prior to induced reactivation and continued through the next 2 days. Vehicle and ACV treated mice were subjected to hyperthermic stress to induce reactivation and 46 hours post stress, ganglia were harvested and processed for whole ganglion IHC to detect HSV protein. The numbers of intact and fragmented neurons were counted (S7 Fig). Consistent with previous findings, fragmentation was observed at this time in a high proportion of the HSV protein positive neurons in vehicle treated mice. ACV treatment dramatically altered this outcome. Fragmented neurons were not detected in any of the acyclovir treated mice at this time (S7 Fig). Although the total number of HSV protein positive neurons was not different between the groups (Student's t-test; $p = 0.12$), the relative distribution between intact and fragmented was significantly different (Fisher's exact test; $p = 0.0004$). This experiment revealed that viral protein expression alone in the latently infected TG environment is not sufficient to drive neuronal fragmentation within the time frame examined.

## Discussion

The pathogenesis of long term HSV infection is still poorly understood. Clinical studies have identified a correlation between neurodegenerative diseases and HSV infection [51], however a mechanism whereby HSV contributes to neurodegeneration remains unclear. Examination of the resolution of HSV reactivation events at the neuronal level is required to understand the impact of periodic reactivation in the nervous system. In this study, we provide evidence that sensory neurons supporting reactivation in vivo undergo apoptosis, visualized as fragmented cellular bodies, which are cleared by Iba1+ cells. Unexpectedly, we did not identify a role for T cell antigen-mediated cytotoxic function in the short term maintenance or direct resolution of reactivation from latency. However, this is consistent with observations that T cells in latently infected TG have an exhausted phenotype and reduced function [52–54].

We validated the use of C57BL/6 mice for in vivo reactivation studies, demonstrating a reactivation frequency similar to more susceptible mouse strains [17, 23]. Previous studies have reported a low frequency of reactivation in C57BL/6 mice [55, 56]. Multiple differences could underlie these distinct outcomes, including inoculum titer, route of infection, method for induction of reactivation, and infectious virus detection protocol. Our study design included two complementary measures of reactivation, the detection of infectious virus as well as localization of viral protein expressing neurons in whole ganglia. To our knowledge, ours is the first study in which entry into the viral lytic cycle was detected and quantified in neurons in the C57BL/6 mouse background, providing additional insight into the changing morphology of neurons undergoing reactivation and the surrounding cellular context. Our study revealed a highly focused immune response to individual neurons undergoing reactivation. A key finding was the detection of viral protein positive fragmented neurons which became the dominant phenotype with time post-reactivation and correlated with declining infectious virus titers in the TG. These fragmented neurons were morphologically consistent with apoptotic bodies and found to contain cleaved caspase-3. In whole ganglia, these fragmented neurons display a striking phenotype and stages of progression from intact to fragmented can be

observed (Fig 3). The rate of progressive destruction of neurons post-reactivation was influenced by viral strain, as evidenced by the higher percent of fragmented neurons at early times (20 h) post-reactivation stimulus of mice infected with HSV-1 strain McKrae compared to 17syn+ (Fisher's Exact test; p<0.0001).

HSV infection of cultured neurons has been shown to result in caspase-3 activation and tau cleavage, which is associated with neurodegeneration [57]. HSV proteins have been shown to inhibit apoptosis during lytic infection (reviewed in: [25]), so it is likely that these same viral proteins also initially block apoptosis as the lytic cycle progresses in the reactivating neuron. The detection of cleaved caspase-3 exclusively in fragmented neurons (rather than intact viral protein positive neurons) was consistent with a blockade to apoptosis early during reactivation allowing for a productive lytic cycle. Interestingly, ACV treatment blocked fragmentation within this time frame. If neurons supporting reactivation progress to the fragmented state, it would be anticipated that cleaved caspase-3 would become detectable and these neurons would be phagocytosed by microglia, which is consistent with our findings here and as observed following neuronal apoptosis in other systems [58]. The presence of both intact and fragmented neurons at the same time may be the result of asynchrony in reactivation initiation/progression. Alternatively, differences in neuronal subtypes, transcription factors levels, chromatin states, and multiplicity of the viral genome, among other variables may affect progression to fragmentation. It also remains possible that some intact neurons do not progress to a fragmented state and can survive reactivation.

In contrast to the *in vivo* setting, in latently infected ganglia that are axotomized and placed in culture (explant; a commonly used reactivation model) neurons undergoing reactivation do not develop a cellular cuff or proceed to fragmentation [22]. Strikingly, the strict containment of virus observed *in vivo* (Figs 3 and 6) is not maintained in explant and evidence of virus spread to neighboring cells can be observed within 24 h [22, 23]. This suggests that control of spread within the TG is linked to an in vivo context-dependent response that promotes cellular containment and ultimately fragmentation of the reactivating neuron. It is worth noting that the immune cell context that evolved within the latently infected ganglia prior to in vivo reactivation or explant would not be different. This suggests that the process of axotomy and explant perturbs the normal (in vivo) response to reactivation, which complicates the interpretation of results obtained in this in vitro explant model [13, 14]. Indeed, results from *ex vivo* studies have been employed to support the conclusion that neurons survive reactivation events [13]. In contrast, the results of the current study reveal a different outcome, namely that the resolution of reactivation *in vivo* involves neuronal destruction.

Neuronal destruction as a result of HSV reactivation is supported by relevant clinical observations in humans, although this is not widely appreciated. The corneas of patients infected with HSV show differences in innervation density and sensitivity to stimuli, compared to uninfected controls [59, 60]. These differences suggest that periodic HSV reactivation leads to axonal damage, which may be the result of neuronal loss. In addition, neurite outgrowth has been detected in skin biopsies of humans with asymptomatic viral shedding [61]. Neurite outgrowth is a response to damage [62] and could be a compensatory mechanism of neighboring neurons following loss of the neuron supporting reactivation. Peng et al. demonstrate that pretreatment of neurons with IL-17c reduces apoptosis related to HSV infection [61], which offers a potential explanation for the high level of specificity for neuronal loss observed in this study. It is possible that IL-17c signaling protects neighboring neurons from death and promotes neurite outgrowth to avoid loss of sensation at the body surface following destruction of the neuron supporting HSV reactivation. Human ganglia infected with HSV have shown signs of chronic inflammation post-mortem and interactions between T cells and neurons, which appear nondestructive, have been identified [12, 63]. Of interest, hypercellular clusters

(nodules) have been identified in human post-mortem samples associated with ganglionic pathology [64–66]. Neuronal destruction and formation of hypercellular nodules post-acute infection with HSV has previously been documented in the mouse model [67] and this study has shown evidence of a similar process occurring post-reactivation (Fig 3). Importantly, elimination of HSV protein positive neurons appeared to be highly selective, since neurons in close proximity to ones supporting HSV reactivation did not display signs of apoptosis or other signs of damage. This likely contributes to observations that the latent pool is not significantly reduced in size despite repeated reactivation events [68]. Loss of 2–3 neurons per reactivation event would not deplete thousands of latently infected sites, even over the lifetime of the host.

The selectivity of neuronal destruction could be due in part to the action of Iba1[+] phagocytic cells, which have previously been shown to play neuroprotective roles following neuronal injury [69]. Expression of Iba1 is upregulated upon macrophage/microglia activation and involved in membrane ruffling and phagocytosis [26]. Iba1 expression was previously shown to increase during acute HSV infection of the TG and brainstem of mice [70]. In our study, Iba1[+] cells were dominant in the cellular cuff surrounding viral protein positive fragmented neurons. We propose that these cells play a key role in eliminating the specific neuron supporting reactivation by phagocytosis while limiting damage and potentially viral spread to surrounding neurons by release of protective and neurotropic factors. This is consistent with previously described mechanisms for clearance of dead neurons in the nervous system [69, 71–74]. Currently we do not know whether the Iba1 expressing cells observed in the TG during reactivation are an expansion of the resident population (Fig 5), infiltrating macrophages from the periphery, or infiltrating microglia from the central nervous system. The movement of central nervous system microglia into the peripheral nervous system in response to peripheral nerve injury and to phagocytose cellular debris has been reported [75]. Further phenotyping of these Iba1[+] cells in the peripheral nervous system and their role in HSV reactivation is ongoing.

In addition to the Iba1[+] cells we identified here, several other cell types have been identified in cellular infiltrates in latently infected ganglia, including CD4[+] and CD8[+] T cells, B cells, and CD11b[+] or F4/80[+] macrophages [16]. Of these, the CD4[+] and CD8[+] T cells have been shown to be important for resolution of the acute infection [44] and CD8[+] T cells directed against an epitope on glycoprotein B are hypothesized to maintain the latent state [76] and there is evidence that this is dependent on MHC class I expression [77]. Further, CXCR3[+] CD8[+] T cells have been shown to increase in the TG and the epithelial surface following a reactivation stimulus and this correlated with a decrease in viral shedding at the surface, noting that outcomes in the TG were not examined [78]. It has also been proposed that regulatory T cells function to suppress CD8[+] T cells and facilitate HSV reactivation [79]. Surprisingly, in our study, loss of CD4 and CD8 coreceptors did not change reactivation with respect to the number of viral protein positive neurons, the production of infectious virus, or the progression of reactivating neurons to a fragmented state at 48 h phs. Furthermore, a cellular response to the neuron supporting HSV reactivation remained intact when CD4/CD8 T cell function was lost. One caveat to these studies is that while *in vivo* antibody treatment was found to deplete the CD4 and CD8 surface receptors to undetectable levels, some T cells were still present in the ganglia (defined as CD3[+], TCRβ[+]). These T cells did not respond to HSV antigens in *in vitro* or *in vivo* assays, but it is possible that T cells could respond to HSV reactivation through antigen-independent mechanisms [80, 81]. A role for T cells in limiting viral spread, potentially through cytokine signaling, is supported by a study by Ramakrishna et al. examining long-term HSV infection in T cell deficient mice [82]. It is important to note that other T cell populations, such as CD3[+]CD4[-]CD8[-], remained present in the ganglia and could also be playing a role in resolution of HSV reactivation [83, 84].

The potential for accrual of damage in the nervous system as a result of HSV reactivation has been proposed to contribute to the development of neurological diseases, such as Alzheimer's Disease [6–8, 85]. Our finding that at least a portion of neurons supporting HSV reactivation do not survive this event and that ACV treatment can alter fragmentation associated with neuronal reactivation contributes to ongoing efforts toward understanding the long-term consequences of HSV infection in the nervous system.

## Materials and methods

### Cells and viruses

Rabbit skin cells (RSC, originally obtained from Dr. B. Roizman at the University of Chicago) were maintained in MEM supplemented with 5% newborn calf serum (Fisher) and incubated at 37˚C in a 5% $CO_2$ incubator. Virus stocks of HSV-1 strain McKrae (originally obtained from Dr. S. Wechsler at Mount Cedar Sinai Medical Center Research Institute) and HSV-1 strain 17syn+ (originally obtained from Dr. C. Preston at MRC Virology Unit in Glasgow, Scotland) were generated by routine propagation on RSC monolayers. Infected RSC were harvested, frozen and thawed three times, and the titer was determined by serial 10-fold dilution plaque assay on RSC monolayers. Following a 2 h incubation period, infected monolayers were overlaid with media containing 1% carboxymethylcellulose and stained with crystal violet 2–3 days later Stocks were aliquoted and stored at -80˚C.

### Inoculation of mice

C57BL/6 (Jackson; 6–8 weeks) and Swiss Webster (Envigo) mice, male or female as indicated, were anesthetized by intraperitoneal injection of sodium pentobarbital (50 mg/kg of body weight), prior to inoculation. A 10 μL drop containing 1 x $10^6$ pfu of 17syn+ or 1 x $10^5$ pfu of McKrae or was placed onto each scarified corneal surface [86]. In this study, infection of C57BL/6 male mice with 2x$10^6$ pfu of HSV-1 strain 17syn+ resulted in 4% (5/114 mice) mortality. Limited mortality was also observed when 17syn+ infection was tested in female C57BL/6 mice (0/5: 0%). Infection of C57BL/6 female mice with 2x$10^5$ pfu of HSV-1 strain McKrae resulted in 41% (96/231) mortality; females were selected for further study of McKrae since mortality was even greater (5/5: 100%) in male C57BL/6 mice infected with McKrae, which has also been reported by others [55, 87].

### In vivo reactivation

Latent HSV was induced to reactivate in vivo by the hyperthermic stress method [17, 48, 86]. In brief, each mouse was placed in a restrainer and suspended in a 42.5–42.8˚C water bath for 10 min. Mice were subsequently towel dried and placed in a 35˚C incubator for 20–30 min to prevent hypothermia. This procedure was performed 3 consecutive times spaced 2.5 hours apart. At the indicated times post the initial stress, TG were removed and processed for the detection of infectious virus or immunohistochemical analysis.

### Detection of reactivated virus

At the indicated times post-hyperthermic stress, TG pairs were removed and homogenized as described previously [86]. The supernatant was plated onto 2 wells of a 6-well plate seeded the day before with RSC. Plates were incubated for 3 hours to allow for viral absorption and then rinsed with fresh media. The next morning, RSC monolayers were overlaid with medium containing 1% carboxy methylcellulose and stained with crystal violet 2–3 days later. To assess recovery using this approach, TG pairs from uninfected and latently infected (pre- and post-

hyperthermic stress) mice were spiked with ~10 pfu 17VP16pLZ; [88] to distinguish input virus from endogenous virus in the infected TG. Viral plaques that expressed β-galactosidase were detected in 100% of TG samples assayed with an average of 3 pfu.

## T cell depletion

Depletion of T cells was achieved by i.p. injection of mice with 300 μg anti-CD4 (GK1.5), 300 μg anti-CD8β (53–5.8), and 400 μg anti-CD8α (2.43) rat monoclonal antibodies or isotype control (BioXcell). Three days after administration of antibodies, mice were given a second dose of antibodies and 1 h later mice underwent hyperthermic stress. Ganglia were harvested at 48 h post-hyperthermic stress and processed for the detection of infectious virus or viral proteins in the whole ganglia. Eyes were maintained in culture for 24 h prior to homogenization to amplify infectious virus, if it was present, and homogenates were then plated on RSC. Depletion of CD4 and CD8 was confirmed at the time of harvest by flow cytometry. Spleens were teased apart by grinding the plunger of a 1 mL syringe against a 70 μm filter. The spleen homogenate was resuspended in 1 mL RBC lysis buffer and held at room temperature for 3 min before quenching in 10 mL tissue culture media. Trigeminal ganglia from two mice were processed as one sample. TG samples were dissociated by gentleMACS Dissociator in 1 mL DMEM and poured over a 70 μm filter. Cells were spun down at 1,400 rpm and pellet was resuspended in HBSS/ 1% FBS and then mixed with Lympholyte-M. Lymphocytes were collected and treated with Fc block (5% mouse serum, 1% BSA in PBS) and incubated with monoclonal antibodies NKp46 (29A1.4) from eBioscience; CD45.2 (104), CD19 (6D5), CD11b (M1/70), CD11c (N418), CD69 (H1.2F3), CD3 (17A2), TCRβ (H57-597), and CD4 (RM4-5) from BioLegend; and CD8 (53–6.7) from Invitrogen. Fluorescence was measured on a FACSCanto flow cytometer (BD Biosciences) and cell populations were analyzed using FlowJo software.

## Antibodies and immunohistochemistry

HSV proteins were detected in whole ganglia as described previously [20]. Primary antibody used was rabbit anti-HSV (Accurate, AXL237) at 1:3,000, or rabbit anti-Iba1 (Abcam; ab178847) at 1:5,000, and secondary antibody used was HRP labeled goat anti-rabbit (Vector) at 1:500. Color development was achieved by exposing ganglia to a 0.1 M Tris (pH 8.2) solution containing 250 μg of diaminobenzidine (Aldrich)/mL and 0.004% $H_2O_2$ for approximately 5 minutes. Ganglia were cleared in glycerol to aid in visualization of the HSV protein positive neurons. TG processed using this method were subsequently rehydrated in PBS, dehydrated in a graded ethanol series, cleared in xylene, and paraffin embedded. Blocks were serially sectioned at 10 μm and consecutive sections were placed on Superfrost Plus slides (Fisher Scientific). Visualization of neurons containing viral proteins in the whole ganglia involved pressing the TG between two glass slides and this handling prior to embedding resulted in the 10 μm sections containing more tissue than a standard 10 μm section prepared by a direct procedure. Deparaffinized and rehydrated sections were briefly exposed to cresyl violet (Sigma). Sections were then rinsed in distilled water, dehydrated, cleared in xylene, and mounted with Permount (Fisher Scientific).

The prior processing by whole ganglia immunohistochemistry was prohibitory to the subsequent detection of some antigens in the sectioned tissue. Additional mice were induced to reactivate by hyperthermic stress and TG were harvested at 34 h phs and directly placed in 4% formaldehyde for 24 h at 4˚C. Latently infected mice were similarly harvested as pre-hyperthermic stress controls in addition to TG from uninfected mice and mock infected mice at 24 and 48 h phs. TG were rinsed in PBS and then dehydrated in a graded ethanol series, cleared in xylene, embedded in paraffin, and sectioned at 10 μm, as described above. Deparaffinized

and rehydrated sections underwent antigen retrieval as follows: slides were placed in Tris-EDTA buffer (pH 9) at 95˚C for 20 minutes and then rested on the bench top for 30 minutes. Rabbit anti-Iba1 (Abcam; ab178847) was used at 1:5,000 and rabbit anti-cleaved caspase-3 (Cell Signaling Technology; #9661) was used at 1:100, followed by incubation with HRP labeled goat anti-rabbit (Vector) at 1:500. Localization of complexes was detected using a solution of diaminobenzidine (Aldrich) or VIP (Vector) according to manufacturer's instructions and sections were counterstained with cresyl violet (Sigma) in some instances. All sections were rinsed in distilled water, dehydrated, cleared in xylene, and mounted with Permount (Fisher Scientific). All slides were viewed under an Olympus BX40 microscope and photographed with Axio-CamHRc (Zeiss). Microscopists were blinded as to time point and viral strain of the sample.

### Ethics statement

All procedures involving animals were approved by the Children's Hospital Institutional Animal Care and Use Committee (IACUC2017-0081) and were in compliance with NIH guidelines. Animals were housed in American Association for Laboratory Animal Care approved quarters.

### Statistical analysis

Statistical analyses were performed using GraphPad Prism software (GraphPad Software, San Diego, CA). $P < 0.05$ is considered significant.

### Replication in vivo

Groups of mice were infected as described above, and at the indicated times post infection, tissues from a minimum of three mice were individually assayed for virus. Tissues were homogenized 1 mL of ice cold MEM, briefly centrifuged, and infectious virus titer was determined by serial 10-fold dilution plaque assay on RSC monolayers. Samples were absorbed for 2 h and overlaid with media containing 1% carboxymethylcellulose. Plates were stained with crystal violet approximately 2–3 days later and the number of plaques were counted.

### Quantification of viral genomes by real time PCR

Isolation and quantification of total viral genomes by real time PCR was performed as detailed previously with some modifications [27].Trigeminal ganglia from three mice maintained for >45 days post-infection were individually processed to determine latent viral genomes. TG were immediately frozen on dry ice and stored at -80˚C until further processing. TG were homogenized in 10mM Tris (pH 8.0), 0.025 mg/mL proteinase K, 0.2% sodium dodecyl sulfate lysis solution and digested overnight at 55˚C. DNA was extracted by standard phenol-chloroform extraction, resuspended in 50 μL 10 mMTris (pH 8.0), and quantified using PicoGreen double-stranded DNA quantification kit (Invitrogen).

A Roche 480 II LightCycler system was used to perform the real-time PCR. Serial dilutions ranging from $10^6$ to $10^0$ copies of plasmid DNA containing the HSV TK locus were used to generate a standard curve. Five microliters containing 50 ng of sample DNA was combined with 15 μL QuantiTest SYBR Green PCR mix (QIAGEN) containing 10 pmol TK primers and run under the conditions previously described [27].

### In vivo reactivation by corneal scarification

Latent HSV was induced to reactivate in vivo by localized stress resulting from corneal scarification, as reported previously [28].

## Inhibition of viral replication by acyclovir

Latently infected mice were treated with acyclovir (50 mg/kg) or vehicle alone twice daily starting at 24 h prior to induced reactivation and continued until the time of sacrifice. Reactivation was induced by hyperthermic stress and TG were harvested and processed for detection of HSV protein by WGIHC at 46 h phs.

## Acute T cell depletion

Depletion of T cells was achieved by i.p. injection of mice with 300 μg anti-CD4 (GK1.5), 300 μg anti-CD8β (53–5.8), and 400 μg anti-CD8α (2.43) rat monoclonal antibodies or isotype control (BioXcell) 1 day post-infection. Mice were infected with $1 \times 10^6$ pfu of 17syn+ and tissues were harvested 7 dpi and processed for infectious virus or HSV protein expression in the whole ganglia.

## Ex vivo antigen stimulation

Dendritic cells were isolated from the spleens of naïve C57BL/6 mice and C57BL/6 mice that were infected for 10 days with 17syn+ based on expression of CD11c by MACS MicroBeads (Miltenyi Biotec), according to manufacturer's instructions. Dendritic cells ($10^5$ cells/well) were left untreated or exposed to UV-inactivated HSV ($10^6$ PFU/well) and incubated with $10^5$ CFSE-labeled CD8 or CD4 T cells isolated from C57BL/6 mice 10 dpi with 17syn+. T cells were first incubated in the presence of 10 μg neutralizing CD4 or CD8 antibody or IgG control antibody (BioXcell). Cultures were maintained for 70 h and cytokine production was determined following a 5 h stimulation with brefeldin A using the Perm/Fix kit (BD Pharmingen) to detect intracellular IFN-γ production.

## In vivo antigen stimulation

Mice latently infected with 17syn+ were injected i.p. with 300 μg anti-CD4 (GK1.5), 300 μg anti-CD8β (53–5.8), and 400 μg anti-CD8α (2.43) rat monoclonal antibodies or isotype control (BioXcell). Three days after administration of antibodies, mice were given a second dose of antibodies and re-infected with $1 \times 10^6$ PFU 17syn+. 24 h post-infection, mice were given 250 μg Brefeldin A and spleens were harvested 6 h post-injection [47]. Spleens were teased apart by grinding the plunger of a 1 mL syringe against a 70 μm filter. The spleen homogenate was resuspended in 1 mL RBC lysis buffer and held at room temperature for 3 min before quenching in 10 mL tissue culture media. Splenocytes were incubated with monoclonal antibodies CD45.2 (104), CD3 (17A2), TCRβ (H57-597), CD4 (RM4-5), and IFNγ (XMG1.2) from BioLegend; and CD8 (53–6.7) from Invitrogen. The presence of bound antibody was evaluated with anti-rat (MRG2b-85) from BioLegend. Fluorescence was measured on a FACSCanto flow cytometer (BD Biosciences) and cell populations were analyzed using FlowJo software. Trigeminal ganglia were processed as previously described [20].

## Supporting information

**S1 Fig. Replication of HSV-1 strains McKrae and 17syn+ in C57BL/6 mice.** Mice were infected on scarified corneas with $1 \times 10^5$ PFU of McKrae or $1 \times 10^6$ PFU of 17syn+ per eye and at the indicated time, tissues from three or more mice were harvested and analyzed for infectious virus titers. (A) Viral replication in the eyes. Infectious virus titers on day 4 post-infection were compared by Student's t-test; $p = 0.0897$. (B) Viral replication in the trigeminal ganglia (TG). Infectious virus titers on day 4 post-infection were compared by Student's t-test; $p = 0.0292$. (C) Survival curve following infection of C57BL/6 female mice with McKrae or

C57BL/6 male mice with 17syn+. Log-rank (Mantel-Cox) p<0.0001. (D) Viral replication in the central nervous system (CNS). High levels of infectious virus were recovered in the CNS of mice with signs of encephalitis (hunching, moribund) while no or low levels of virus were recovered from mice lacking such signs.
(TIF)

**S2 Fig. Frequency of intact HSV expressing neurons is inversely related to frequency of fragmented HSV expressing neurons.** The average number of intact and fragmented neurons in ganglia that contained positive neurons (+veTG) as determined in Fig 1 (minimum of 10 ganglia per time point) was plotted at each time point post-reactivation induction (hrs pi). A negative linear regression was determined for neurons of the intact phenotype and a positive linear regression was found for neurons of the fragmented phenotype. $R^2$ values are given on the graph for each relationship.
(TIF)

**S3 Fig. Neuronal fragmentation is neither mouse strain nor reactivation stressor specific.** The number of intact and fragmented neurons and examples of individual fragmented neurons in trigeminal ganglia of Swiss Webster mice following reactivation induced by hyperthermic stress (A,B) or in mice following reactivation induced by corneal scarification (C,D) are shown. Bars in A and C indicate the number of intact or fragmented viral protein positive neurons in 10 ganglia/group at each time post-induction. Examples of HSV viral protein positive fragmented neurons are shown in B and D.
(TIF)

**S4 Fig. Acute infection in CD4 and CD8 depleted mice.** Mice were infected on scarified corneas with 1 x10$^6$ PFU of 17syn+ per eye and 1 day later treated with anti- CD4/CD8 depleting/neutralizing antibodies or control IgG. On day 7 pi, tissues from 3 mice in each group were harvested and analyzed for infectious virus titers or viral protein expression. (A) Viral titers in the eyes and TG from control and anti-CD4/CD8 treated mice. Anti-CD4/CD8 treatment resulted in significantly higher viral titers in both the eyes (Student's t-test; p = 0.019) and TG (Student's t-test; p = 0.025). Line indicates average of 3 samples. (B and C) Viral protein expression (brown DAB reaction product) in ganglia from control (B) and anti-CD4/CD8 (C) treated mice. Blue arrows indicate neurons expressing viral proteins. Black arrows indicate viral protein expression in cells lining the axonal tracts. Low power and high power views are shown to emphasize the striking differences in number of infected neurons in TG from anti-CD4/CD8 treated mice. A fragmented neuron is shown at higher power (boxed inset).
(TIF)

**S5 Fig. Functional neutralization of CD4 and CD8 T cells in vitro.** Control T cells (C-T cells) and T cells from HSV infected mice (HSV-T cells) were harvested and incubated with anti-CD4, anti-CD8, or control IgG and presented with HSV1 antigen from dendritic cells (HSV) or unexposed control dendritic cells (C) as detailed in Methods. Production of intracellular IFNγ was measured 70 h later. Addition of both anti-CD4 and anti-CD8 significantly reduced T cell IFNγ production compared to control IgG treated T cells from HSV infected mice presented with HSV antigen exposed dendritic cells. The percent cells producing IFNγ in cultures treated with anti-CD4/CD8 was not significantly different from background levels detected in control conditions. One-way ANOVA with Tukey's multiple comparison test: $^*$ = p<0.05; $^{**}$ = p<0.01; $^{***}$ = p<0.001; $^{****}$ = p<0.0001.
(TIF)

**S6 Fig. Anti-CD4 and anti-CD8 antibodies prevent IFNγ production in response to HSV in vivo.** Naïve mice [HSV(-)] and mice latently infected with HSV-1 strain 17syn+ [HSV(+)] were treated with control IgG or anti-CD4 and anti-CD8 depleting antibodies. Mice were given a second dose of antibodies 3 days later and the latently infected mice were re-infected with 1 x10$^6$ PFU of 17syn+. Brefeldin A (BFA) was given i.p. 24 post-infection and tissues were harvested 6 hours post-injection. Values from naïve mice are plotted in parallel to determine background fluorescent levels. (A) Schematic representation of experimental approach. (B) Flow cytometric analysis of IFNγ expression in the CD3+ TCRβ+ population. (C) Percent of IFNγ producing cells in the CD3+ TCRβ+ population as determined by flow cytometric analysis. One-way ANOVA with Tukey's multiple comparison test; ** = p<0.01. (D) Intracellular IFNγ was detected in whole ganglia (brown; DAB reaction product). Arrows indicate cells positive for IFNγ.
(TIF)

**S7 Fig. Inhibition of viral DNA replication during reactivation blocks neuronal fragmentation.** Mice latently infected with 17syn+ were treated with ACV (50 mg/kg, ip, 2x daily) or saline (mock) control (n = 5 mice/group) starting one day prior to hyperthermic stress and continued until the time of sacrifice. At 46 h phs, ganglia were removed and HSV proteins detected by WGIHC. (A) Bars represent the number of intact and fragmented viral protein expressing neurons (n = 10 TG per group). Fragmented neurons were not detected in the ACV treated group. *ACV treated and untreated groups are different, p = 0.0004, Fishers exact test. (B) Photomicrographs of fragmented, HSV protein positive neurons detected in mock treated mice. (C) Photomicrographs of intact, HSV protein positive neurons detected in ACV treated mice.
(TIF)

## Acknowledgments

We thank Kristin Lampe for expert technical assistance.

## Author Contributions

**Conceptualization:** Jessica R. Doll, Richard L. Thompson, Nancy M. Sawtell.

**Data curation:** Jessica R. Doll, Nancy M. Sawtell.

**Formal analysis:** Jessica R. Doll, Kasper Hoebe, Richard L. Thompson, Nancy M. Sawtell.

**Funding acquisition:** Nancy M. Sawtell.

**Investigation:** Jessica R. Doll, Nancy M. Sawtell.

**Methodology:** Jessica R. Doll, Kasper Hoebe, Nancy M. Sawtell.

**Project administration:** Jessica R. Doll, Nancy M. Sawtell.

**Resources:** Kasper Hoebe, Richard L. Thompson, Nancy M. Sawtell.

**Supervision:** Jessica R. Doll, Nancy M. Sawtell.

**Validation:** Jessica R. Doll, Nancy M. Sawtell.

**Visualization:** Jessica R. Doll, Nancy M. Sawtell.

**Writing – original draft:** Jessica R. Doll, Nancy M. Sawtell.

**Writing – review & editing:** Jessica R. Doll, Kasper Hoebe, Richard L. Thompson, Nancy M. Sawtell.

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
