## [Decision Letter · Decision Letter 0]

14 Aug 2019

Dear Dr Sawtell,

Thank you very much for submitting your manuscript "Resolution of herpes simplex virus reactivation in vivo results in neuronal destruction" (PPATHOGENS-D-19-01139) for review by PLOS Pathogens. Your manuscript was fully evaluated at the editorial level and by independent peer reviewers. The reviewers appreciated the attention to an important problem, and that the study has the potential to make a substantial contribution to the literature. There are however some substantial concerns about the manuscript as it currently stands. These issues must be addressed before we would be willing to consider a revised version of your study. We cannot, of course, promise publication at that time.

All reviewers acknowledge the high quality of your experimental work and that your study addresses a very important long open standing topic. However, particularly reviewer 1 and reviewer 3 feel that there are significant gaps that need to be filled.

All reviewers ask for more information on the reactivation state and the fate of HSV positive neurons, and the potential temporal relationship of neuronal apoptosis and nodules. Furthermore, reviewer 1 and reviewer 3 request a further characterization of the immune cells surrounding the HSV positive neurons. In particular, I would like to highlight the comments of reviewer 3 on the T cell depletion experiments. She/he questions whether it has been sufficiently investigated which cell types have been depleted, and whether there may be a remaining T cell activity that might contribute to the resolution of the reactivated HSV infection.

The suggestion of reviewer 1 for an additional animal model seems to be beyond the scope of the present study. However, we agree with reviewer 3 that further references to similarities and difference to the mechanisms of HSV reactivation in patients and in other animal models with particular reference to the role of T cells in the discussion would make the study more accessible to a broader readership.

All reviewers have further specific suggestions to improve the clarity of the manuscript.

We therefore ask you to modify the manuscript according to the review recommendations before we can consider your manuscript for acceptance. Your revisions should address the specific points made by each reviewer.

(1) A letter containing a detailed list of your responses to the review comments and a description of the changes you have made in the manuscript. Please note while forming your response, if your article is accepted, you may have the opportunity to make the peer review history publicly available. The record will include editor decision letters (with reviews) and your responses to reviewer comments. If eligible, we will contact you to opt in or out.

(2) Two versions of the manuscript: one with either highlights or tracked changes denoting where the text has been changed; the other a clean version (uploaded as the manuscript file).

Additionally, to enhance the reproducibility of your results, PLOS recommends that you deposit your laboratory protocols in protocols.io, where a protocol can be assigned its own identifier (DOI) such that it can be cited independently in the future. For instructions see http://journals.plos.org/plospathogens/s/submission-guidelines#loc-materials-and-methods

We hope to receive your revised manuscript within 60 days. If you anticipate any delay in its return, we ask that you let us know the expected resubmission date by replying to this email. Revised manuscripts received beyond 60 days may require evaluation and peer review similar to that applied to newly submitted manuscripts.

[LINK]

Sincerely,

Beate Sodeik

Guest Editor

PLOS Pathogens

Klaus Früh

Section Editor

PLOS Pathogens

Kasturi Haldar

Editor-in-Chief

PLOS Pathogens

orcid.org/0000-0001-5065-158X

Grant McFadden

Editor-in-Chief

PLOS Pathogens

orcid.org/0000-0002-2556-3526

Reviewer's Responses to Questions

**Part I - Summary**

Reviewer #1: Report on the manuscript by Doll et al., entitled “Resolution of herpes simplex virus reactivation in vivo results in neuronal destruction”

This report addresses a very important and relevant question: what is the fate of the neuron upon of herpes simplex virus type 1 (HSV-1) reactivation. To answer this question the authors infect C57BL/6 mice through the cornea with two HSV-1 strains, induce reactivation by hyperthermic stress 45 days post-infection and determine the number and fate of reactivated neurons using staining techniques, quantitative PCR and plaque assays. They define latency by the lack of detection of viral protein and infectious viral particles. Reactivation is defined here by the detection of viral protein and/or infectious viral particles. The authors infect female and male mice with the McKrae and 17syn+ strains, respectively and reach similar conclusions despite the differences in strain and gender.

Initially, they authors perform a thorough characterization of the viral replication kinetics, the time of latency establishment and reactivation. They find out that the number of neurons positive for HSV-1 antigens reaches a peak around 24 hours post-reactivation and then decreases. This correlates with the viral titres in TG, which are highest between 20-24 hours post-reactivation. Interestingly, some neurons have an intact morphology whereas others seem fragmented or damaged and are positive for the apoptotic marker caspase 3. The percentage of such neurons, but not their total number, increases with time post-reactivation. The authors find that these neurons are surrounded by infiltrating cells, many of them expressing Iba1 and considered by the authors to be microglia. They also find nodules in spaces possibly occupied previously by apoptotic neurons. Since T cells are thought to be relevant on inhibiting HSV-1 reactivation the authors deplete them prior to hyperthermic stress. Surprisingly, the data suggests that T cells do not play a role in controlling HSV-1 reactivation in this model.

The main findings of this study are: (i) the discovery of apoptotic neurons upon HSV-1 reactivation; (ii) the formation of nodules surrounded by Iba1+ cells, suggesting that they are involved in the clearance of reactivated, apoptotic neurons; (iii) CD4 and CD8 positive T cells do not play a key role in controlling HSV-1 reactivation in the mouse model despite previous results suggesting the contrary.

The manuscript is very well written and clear, most figures are of high quality. The experiments are performed well, including the relevant controls. Most conclusions are supported by the data, although others are not.

Reviewer #2: The authors are known for their painstaking and careful morphologic work, leadership in the hyperthermic stress model, and have a track record of influencing the field. A plus is that they study two different viral strains. The authors nicely check for latency at day 30 vs 45 and for differences in latent HSV DNA load between models. McKrae, even at a lower inoculum, replicated to higher titers in eye and TG at day 4 and resulted in more mouse death. Data are presented for one sex for each viral strain, with a note that McKrae was even more pathogenic in males.

Reviewer #3: In the manuscript entitled “resolution of herpes simplex virus reactivation in vivo results in neuronal destruction”, Doll et al. present the results of a very extensive analysis of the fate of latently infected murine trigeminal ganglia neurons that have been induced to reactivate HSV-1 by hyperthermic stress in vivo. For this study, the authors use two HSV-1 strains that are efficient reactivators in the mouse model (17+ and McCrae) and examine reactivation by two classic criteria: the ability to detect infectious virus and the ability to detect viral antigen in individual neurons by immunohistochemistry. In order to address the question of neuronal fate after reactivation, ganglia are examined at several time points after thermal stress: 20, 34 and 48 hrs. Key findings of this study are: 1) reactivation can be detected by 20 hours post stress and some of the antigen positive neurons exhibit fragmentation (an indication of possible apoptosis), 2) the number of neurons exhibiting fragmentation increases at the later time points, 3) the fragmented neurons show evidence of caspase 3 cleavage, indicating they are undergoing apoptosis, 4) latently infected ganglia show increases in activated microglia, but reactivating ganglia show evidence of clustering of these microglia around antigen positive cells, but especially around fragmented cells, suggesting that these cells may play a role in clearance of these cells, and 5) blocking of HSV-1 antigen-specific T cells does not alter the number of reactivating neurons, nor their fate.

This is a highly significant study as it attempts to address the age-old question regarding whether neurons that reactivate HSV latent infection die as a result of the reactivation, or whether they can survive. As the authors point out, there are publications that support both possibilities. In particular, explant-induced reactivation does not tend to show robust evidence for neuronal death as a direct result of reactivation. This is the first study to take this question on looking at reactivation in vivo. The study is very well-designed and carefully performed. Because reactivation is a relatively rare event, large cohorts of mice were examined to allow statistically defendable conclusions to be drawn. This is an outstanding study and one that will be highly cited and have a large impact on the field. There are only a few relatively minor points that should be addressed for precision and clarity.

**Part II – Major Issues: Key Experiments Required for Acceptance**

Reviewer #1: 1. The main objective of this manuscript is to determine the fate of the neurons in which HSV-1 reactivated. This is a very difficult task due to the low number of neurons in which reactivation is detected. This is not a criticism to the authors, it is more a description of the challenge faced by them. The authors find both neurons that have a normal morphology and others that look “fragmented” upon reactivation. Although in lines 452-454 the authors suggest that in some neurons the viral cycle will progress and apoptosis will be inhibited, in other sections of the paper such as lines 463-465 it seems that the authors promote the idea that fragmentation and apoptosis of the reactivated neuron is the main mechanism to control reactivation. Even the title of the manuscript suggests this. However, I do not think that this can be concluded from the current data.

Similarly, based on data shown in Figure 2, the authors conclude that there is a transition from non-fragmented to fragmented neurons because the percentage of the latter increases with time. However, the total number of fragmented neurons does not increase as one would expect if most non-fragmented neurons become apoptotic. Since the authors also found the presence of nodules and suggest that these appear in places in which apoptotic neurons were located one would also expect that the number of nodules increases with time reaching a level similar to that of reactivated neurons. The authors show that the number of nodules increases with time (Table 3) but is difficult to compare these numbers with the numbers of reactivated neurons shown in Figure 2 since there is no information on the number of ganglia used for this figure. However, the low number of nodules shown in Table 3 suggests that many of the reactivated neurons did not become nodules. This would suggest that the fate of many reactivated neurons is not to undergo apoptosis and that reactivation is controlled by an alternative mechanism.

The authors base their conclusions on a systematic analysis to identify neurons positive to HSV-1 antigen. However, it is possible that HSV-1 reactivated in more neurons without generating enough viral antigens to be detected by antibody staining. The use of hybridization techniques to identify HSV-1 genome and transcripts will detect such neurons allowing the quantification of the percentage of reactivating and apoptotic neurons. This will shed some light on how common the described phenotype is.

2. As indicated in lines 254-257, several cell types are involved in removing cellular debris in peripheral ganglia. However, the authors focus on microglia, a cell type more commonly found in the CNS than in peripheral ganglia (although they can also enter the PNS, i.e., Smith et al., 2019, PLoS Biology and other reports). It may help the reader who considers them as CNS-restricted cells if this is explained in the manuscript. A possible alternative to the authors’ conclusions is that the Iba1+ cells are not microglia but infiltrating macrophages since they also express this protein (Ohsawa et al., 2004 J Neurochemistry Feb;88(4):844-56; Sasaki et al., Biochem Biophys Res Commun. 2001 Aug 17;286(2):292-7, among others). P2Y12 is expressed in microglia and not macrophages and is currently used as a marker to distinguish between these two cell types. Moreover, its expression is increased upon microglia activation. The authors should use this marker to clearly demonstrate that microglia and no macrophages surround the reactivating neurons. Are there other cells, not Iba1+, surrounding the neurons during reactivation? If so, which ones?

One interesting question arising from this work is whether the Iba1+ cells are there to clear the apoptotic neuron or whether they are involved in control of reactivation or in the induction of apoptosis, since non-fragmented neurons are also surrounded by these cells, although to a lesser extent (Figure 4 E, F). Also, in Figure 5 the Iba1+ cells have a different morphology in the mock and latently infected ganglia. In certain areas of the latent ganglion there is more Iba1 expression and the Iba1+ cells have a rounded morphology indicative of activation, whereas in others they look as non-activated cells. A quantification of both cellular phenotypes would facilitate asserting the relevance of these observations. According to Figure 5B, the Iba1+ cells also surround the neurons before reactivation and cover nodules where presumably a neuron died during lytic infection. It would be helpful to detect the viral genome by FISH to determine whether there are activated Iba1+ cells already surrounding the latently infected neurons. Also, detection of HSV transcripts by RNA-scope or similar technique will inform whether reactivation already started in some of these neurons but did not progress to production of protein and infectious viral particles. This would indicate whether the Iba1+ cells participate in initial control of reactivation. Once the authors convincingly prove that the cells are microglia or macrophages it will be very interesting to determine their role in neuronal fate upon HSV-1 reactivation by depleting them prior to reactivation. There are protocols available in the literature to deplete either cell population in the mouse.

3. Previous studies did not show efficient reactivation in C57BL/6 mice, contrary to the findings in this report. The authors discuss nicely the possible reasons for these discrepancies. Here, the analysis of protein expression in neurons and the presence of infectious virus in TG demonstrate reactivation. But there is no data showing the presence of HSV-1 in the eye or tear film upon reactivation. It is possible that C57BL/6 mice are particularly efficient in controlling HSV-1 reactivation and that a different phenotype may be observed in other mouse models in which reactivation is more efficient. Although I am aware of the difficulty of setting up a similar model with a new mouse strain, performing such experiments will allow determining whether the Observations reported here are mouse-strain specific or a general phenomenon. This will also provide information of the mechanism(s) leading to control of reactivation.

Reviewer #2: 1. Overall there were relatively few HSV IHC (+) cell outlines in the entire study, emphasizing careful microscopy. Were the microscopists evaluating from intact vs fragmented HSV protein (+) cell outlines (Fig. 2) blinded as to time point or viral strain? Similarly for Fig. 1C,D were the microscopists blinded for time point? The same blinding query extends to additional data sets and observations.

2. I am concerned about the interpretation of the data about the temporal progression of inflammatory cuffs surrounding HSV antigen (+) zones over time discussed in lines 291-327. The authors note that at their early time point (20 hours after hyperthermia) there is a mix of cuffed and non-cuffed neuron-like cells, but that by later time points (48 hours) all of the HSV IHC (+) areas (interpreted as neurons) have an inflammatory cuff. Of concern, the absolute number of HSV IHC (+) areas at the late time points is very small, such that the ratio of cuffed to non-cuffed HSV IHC (+) areas is unlikely to be measured with much precision. Between McKrae and 17syn+, there were only 6 HSV IHC (+) areas observed at 48 hours (Tables 1 and 2). Lines 437-438 in the Discussion also concern the proportion of fragmented neurons and this again is based on a very limited number of observations. It is not clear if the nodules discussed in lines 328-335 are really areas of inflammation localized to (now cleared) reactivated neurons: were these nodules also HSV IHC (+),or just clumps of Ib1a1+ cells? ? Can we see an example in a micrograph? An average of one such nodule was seen per TG (Table 3) and the legend line 919 indicates that these were “likely” sites of previous neurons. Given the small numbers of these structures and the incomplete documentation that these nodules are sites of cleared neurons, I think these findings should be de emphasized.

3. One of the key conclusions of the paper is that T cells are not involved in the morphologic disruption of HSV IHC (+) areas that probably represented dying reactivated neurons. This is a potentially paradigm-breaking finding in this field, as both prior human and mouse data indicate that HSV-specific T cells form halos around latently infected neurons, and depletion of T cells in various reactivation models leads to increased HSV replication. It would help to build the case for this statement by staining the cell halos around HSV IHC (+) cell areas (as in Fig. 4 F) for T cell markers, in addition to the microglial marker. The simple presence of T cells could not imply a causal role in viral shutdown, but their convincing absence would build the case.

4. The main technology used was administration of mAbs against CD4, CD8 alpha, and CD8 beta. In preliminary experiments (Sup Fig 2), administration of anti-CD4 and anti-CD8 prior to acute infection leads to increased recovery of HSV at day 7 from eye and brain after eye inoculation. The authors then shift to administering the mAbs 3 days and 1 hour prior to hyperthermic stress in latently infected mice. Fig 7 shows that cells that are CD3(+) TCRbeta positive are still present in spleen and TG at 48 hours after the hyperthermic stress. Thus, the T cells are not depleted: at least some cells are still present, but the CD4 and CD8 epitopes are interpreted as masked by the overwhelming amounts of mAb administered previously to the mice. What we don’t know about in enough detail is what these residual cells are: are they CD4negCD8neg T-like cells (as in some gamma delta T cells or NKT cells) or are they traditional CD4+CD8- or CD4-CD8+ T cells with masked surface CD4/CD8 co-receptors as hypothesized by the authors. We also don’t; know if they are functionally blocked by the mAbs.

The human literature contains well described CD8 co-receptor independent T cell clonotypes: for these T cells, the TCR itself can bind strongly to peptide-MHC, and CD8 co-receptor binding to MHC class I is not required for T cell signaling and function. Thus, the absence of stain-accessible CD8 does not mean that residual traditional CD4-CD8+ T cells, if present, are non-functional.

The authors try to address this issue by incubating immune splenocytes with HSV-charged DC as APC in the presence or absence of mAbs (Fig. 7). In these expts, the mAbs are added in vitro; it would have been better to use responder splenocytes from animals that had previously been given mAbs in vivo. They use inactivated HSV to charge the DC, and as expected, they see relatively little CD8 reactivity (little decrement on adding blocking only anti-CD8). DC cross-presentation of inactivated antigen can occur but is typically very challenging to detect. In the data shown, the addition of anti CD8 alone leads to lonely a partial decrement in T cell activation, while anti-CD4 leads to better blockade that is not increased by adding anti CD8 to the anti CD4.

Thus, we are not really presented data from a system in which anti-CD8 could even potentially block HSV-specific CD8 T cell responses. It would be more convincing to isolate tetramer (+) CD8 T cells specific for the well-known HSV-1 gB immunodominant CD8 epitope and test if the anti-CD8 mabs would block their activation in response to peptide-pulsed or HSV-infected APC, which are more traditional APC for CD8 T cells. Regardless, the use of high doses of candidate blocking mAb in vitro is less convincing than testing responder cells recovered from mAb-treated mice. The finding of neuron fragmentation in the mAb-treated animals given hyperthermic stress could be due to residual CD8 T cell activity that was not depleted/blocked.

5. In human medicine, there is no evidence of neuron drop-out in post-mortem exams of HSV (+) vs. HSV (-) ganglia. Many neurons in TG are DNA (+) or LAT (+) or both even in very elderly people, consistent with long term survival of at least some infected neurons. There is also no clinical evidence of anaesthesia in lip or genital areas supplied by HSV-infected ganglia that might occur if there was significant loss of neurons. The Discussion should acknowledge the differences in pathogenesis and the difficulty of drawing conclusions in humans.

6. Neuron dropout/death in the mouse model was associated with a halo of glial-like cells. Since we don’t know if neuron dropout occurs in humans, the significance of this finding to humans is also unknown. Both host species share the finding of halos or rims of HSV-specific T cells (including HSV tetramer (+) CD8 T cells with markers of cytolytic effector function) that surround HSV-infected ganglionic neurons. Thus there is the potential for T cell recognition of HSV proteins (either in reactivating neurons or surrounding satellite glial cells (SGC)), and either cytolytic or non-cytolytic T cell effector functions influencing the reactivating neuron. In vitro, upon explant, neutralization of CD8 or IFN-g leads to increased lytic HSV reactivation of mouse neurons.

Ghiasi et al. have also re-addressed this issue with findings that CD8+ DC may account for some anti-CD8 findings, an issue that should be discussed. This issue of whether or not acquired T cells contribute to control of reactivation in vivo is of great translational concern, as therapeutic vaccines for HSV are largely T cell oriented and current thinking is to try to get antigen-specific T cells into ganglia.

Reviewer #3: None

**Part III – Minor Issues: Editorial and Data Presentation Modifications**

Reviewer #1: 1. Mouse models are essential to understand HSV-1 pathogenesis. At the same time, some conclusions or observations obtained using mice do not apply to humans. In this report reactivation is rapidly controlled and whether infectious virus is present in the eye or tear film after reactivation is not indicated. Do the authors detect infectious virus in the eye upon reactivation? The authors indicate in lines 99-100 that “inflammatory foci have been detected in HSV infected human ganglia post-mortem, independent of viral protein expression (11, 12)”. They also mention in lines 474-476 observations supporting death of neurons after HSV-1 reactivation in humans. However, it is also known that HSV-1 reactivates asymptomatically in humans quite frequently, and infectious virus is detected in the innervated mucosa. Moreover, HSV-2 reactivation in human sacral DRG is associated with increased neurite outgrowth and not with neuronal death (Peng et al., J Exp Med 2017, doi: 10.1084/jem.20160581). The data presented here provides support for the death of neurons upon reactivation but, to my knowledge, it is not clear at present what the percentage of reactivated neurons that undergo apoptosis upon reactivation in humans is. Is there a “reactivation threshold” that leads to neuronal apoptosis? Or does this also occur during asymptomatic reactivation? Although it is beyond the scope of this work to obtain answers to these questions, it makes sense to take all these previous observations into account and discuss the relevance in humans of the interesting findings reported here.

2. Figure 4: The authors show that 3 out of 3 fragmented neurons are positive for caspase 3 whereas non-fragmented ones are negative. Although the data looks convincing it would be even more so if the n is increased. The caspase 3+ neuron is surrounded by Iba1+ cells. Was this observed in the other apoptotic neurons found?

3. In some experiments, such as the ones in Tables 2 and 3, Figures 2 and 4, it is not clear the number of ganglia used. Also, what is the variation between different ganglia?

4. Lines 432-435. Maybe I do not understand what the authors imply, but I do not agree with the current statement. The number of HSV-1 positive neurons associated with inflammatory cuff does not increase with time post-reactivation. Tables 1 and 2 show more neurons associated with cuff at 20 hours than later. The percentage is higher but this may be due to the fact that the number of events is highly reduced with time.

5. Some references are not inserted with endnote or similar software. Instead, the surname of first author and year are included.

Reviewer #2: 1. Was there a sex difference for 17syn+?

2. There are some issues with the Fig. 4 legend. I don’t understand which of the 4 X marks in B correspond to which of the 3 X marked neuros in D. While I can intuit what E is, it is not called out in the legend. Similar for F. G and H appear to be lower power shots of a ganglia with an iba1(+) infiltrate around an HSV (+) cell but this is not called out in the legend. It is called out in the text around line 248 but better to also mention in legend.

3. The morphologic findings in Fig. 5E seem to represent a single finding in a single ganglia, and as such I question whether they should be presented as a characteristic of post-hyperthermic stress samples. In addition, the interpretation in the results lines 289-290 is conjecture, with no visualization of egressing virus, so I suggest this either be relegated to Discussion, or better, dropped. Similarly the line 995-996 statement that the cell cluster might be the site of previously infected neuron in a pre-stress latent ganglia is not supported by data and should be dropped.

4. They key Fig. 7B, C data showing no difference in reactivation parameters in mAB-treated mice is a little hard to interpret. What virus strain was used and what time point was studied? Of note the number of HSV IHC (+) cells was far lower in Fig 7C untreated animals (1-2 per ganglia) than in the Fig. 1 C, D lead-in experiments: why is this?

5. The p value in line 447 appears to be based on differences in McKrae vs 17syn+ between Tables 1 and 2 at 20 hours. The mention of 24 hours in line 445 is confusing as 24 hours is not mentioned elsewhere. The numbers of fragmented outlines for McKrae at 20 hours (3, table 1) and for 17syn+ (0, table 2) are very low absolute numbers, such that I question use of stats and drawing conclusions on such small numbers of observations.

6. As a minor point in Sup 1D were brain dissections and titers done on day 10 in the 17syn+ mice?

Reviewer #3: 1) The largest issue is that I believe that the manuscript gives the impression that HSV-1 reactivation itself induces the apoptosis that is being observed and arguing that immune factors (microglia and/or cytokines are not involved). The authors do elude to this possibility in the discussion, but I think it is important enough a point that it be more explicitly stated in the body of the manuscript whether this is discussed as well (for example, lines 264 – 265).

2) line 184. It is stated that “less than 2 fold variation” in establishment is detected. This does not seem correct as the variation for each virus seems to be greater than that (lines 186 – 187)?

3) Related to point 1 above, the way that the title, and conclusions in the abstract and throughout the text of the manuscript read, the authors seem to imply that all reactivating (or antigen-positive) neurons die by apoptosis. While I am convinced from the data presented that a majority of them seem to, the data do not exclude the (likely) possibility that some of them do not. This should be explicitly stated and discussed for completeness).

4) Related to point 4, the sentence on line 528 should be modified to state that “at least some neurons supporting HSV-1 reactivation are destroyed by the event”.

5) Lines 531 – 532. Probably should tine down the statement that suggests CD4+ and CD8+ T cells play no role in maintaining the latent state. As discussed in the paragraph above this statement, they may well play a role in cytokine signaling.

PLOS authors have the option to publish the peer review history of their article (what does this mean?). If published, this will include your full peer review and any attached files.

Reviewer #1: No

Reviewer #2: No

Reviewer #3: No

---

## [Decision Letter · Decision Letter 1]

26 Dec 2019

Dear Dr Sawtell,

We are pleased to inform that your manuscript, "Resolution of herpes simplex virus reactivation in vivo results in neuronal destruction", has been editorially accepted for publication at PLOS Pathogens. 

Before your manuscript can be formally accepted and sent to production, you will need to complete our formatting changes, which you will receive by email within a week. Please note that your manuscript will not be scheduled for publication until you have made the required changes.

IMPORTANT NOTES

(1) Please note, once your paper is accepted, an uncorrected proof of your manuscript will be published online ahead of the final version, unless you’ve already opted out via the online submission form. If, for any reason, you do not want an earlier version of your manuscript published online or are unsure if you have already indicated as such, please let the journal staff know immediately at plospathogens@plos.org.

(2) Copyediting and Proofreading: The corresponding author will receive a typeset proof for review, to ensure errors have not been introduced during production. Please review the PDF proof of your manuscript carefully, as this is the last chance to correct any errors. Please note that major changes, or those which affect the scientific understanding of the work, will likely cause delays to the publication date of your manuscript. 

(3) Appropriate Figure Files: Please remove all name and figure # text from your figure files. Please also take this time to check that your figures are of high resolution, which will improve the readbility of your figures and help expedite your manuscript's publication. Please note that figures must have been originally created at 300dpi or higher. Do not manually increase the resolution of your files. For instructions on how to properly obtain high quality images, please review our Figure Guidelines, with examples at: http://journals.plos.org/plospathogens/s/figures.

(4) Striking Image: Please upload a striking still image to accompany your article if one is available (you can include a new image or an existing one from within your manuscript). Should your paper be accepted, this image will be considered for our monthly issue image and may also appear on our website to feature your article. Please upload this as a separate file, selecting "striking image" as the file type upon upload. Please also include a separate "Other" file with a caption, including credits and any potential copyright information. Please do not include the caption in the main article file. If your image is from someone other than yourself, please ensure that the artist has read and agreed to the terms and conditions of the Creative Commons Attribution License at http://journals.plos.org/plospathogens/s/content-license. Please note that PLOS cannot publish copyrighted images.

(5) Press Release or Related Media: If your institution or institutions have a press office, please notify them about your upcoming paper at this point, to enable them to help maximize its impact. If they will be preparing press materials for this manuscript, please inform our press team in advance at plospathogens@plos.org as soon as possible. We ask that you contact us within one week to plan ahead of our fast Production schedule. If you need to know your paper's publication date for related media purposes, you must coordinate with our press team, and your manuscript will remain under a strict press embargo until the publication date and time. This means an early version of your manuscript will not be published ahead of your final version. 

(6)  PLOS requires an ORCID iD for all corresponding authors on papers submitted after December 6th, 2016. Please ensure that you have an ORCID iD and that it is validated in Editorial Manager.  To do this, go to ‘Update my Information’ (in the upper left-hand corner of the main menu), and click on the Fetch/Validate link next to the ORCID field.  This will take you to the ORCID site and allow you to create a new iD or authenticate a pre-existing iD in Editorial Manager

(7) Update your Profile Information: Now that your manuscript has been provisionally accepted, please log into Editorial Manager and update your profile, if needed. Go to https://www.editorialmanager.com/ppathogens, log in, and click on the "Update My Information" link at the top of the page. Please update your user information to ensure an efficient production and billing process. 

(8) LaTeX users only: Our staff will ask you to upload a TEX file in addition to the PDF before the paper can be sent to typesetting, so please carefully review our Latex Guidelines http://journals.plos.org/plospathogens/s/latex in the meantime.

(9) If you have associated protocols in protocols.io, please ensure that you make them public before publication to guarantee immediate access to the methodological details.

Best regards,

Beate Sodeik

Guest Editor

PLOS Pathogens

Klaus Früh

Section Editor

PLOS Pathogens

Kasturi Haldar

Editor-in-Chief

PLOS Pathogens

orcid.org/0000-0001-5065-158X

Grant McFadden

Editor-in-Chief

PLOS Pathogens

orcid.org/0000-0002-2556-3526

Reviewer Comments (if any, and for reference):

Reviewer's Responses to Questions

**Part I - Summary**

Reviewer #1: The authors have addressed most of my concerns. The incorporated modifications, including new experiments and changes in the text, clearly improved the manuscript. The results provide substantial evidence for the authors' conclusions. I believe that this report will contribute to the understanding of the neuronal fate upon HSV-1 reactivation and it will spark scientific discussions and trigger more investigations in this exciting field.

Reviewer #3: In the manuscript entitled “resolution of herpes simplex virus reactivation in vivo results in neuronal destruction”, Doll et al. present the results of a very extensive analysis of the fate of latently infected murine trigeminal ganglia neurons that have been induced to reactivate HSV-1 by hyperthermic stress in vivo. For this study, the authors use two HSV-1 strains that are efficient reactivators in the mouse model (17+ and McCrae) and examine reactivation by two classic criteria: the ability to detect infectious virus and the ability to detect viral antigen in individual neurons by immunohistochemistry. In order to address the question of neuronal fate after reactivation, ganglia are examined at several time points after thermal stress: 20, 34 and 48 hrs. Key findings of this study are: 1) reactivation can be detected by 20 hours post stress and some of the antigen positive neurons exhibit fragmentation (an indication of possible apoptosis), 2) the number of neurons exhibiting fragmentation increases at the later time points, 3) the fragmented neurons show evidence of caspase 3 cleavage, indicating they are undergoing apoptosis, 4) latently infected ganglia show increases in activated microglia, but reactivating ganglia show evidence of clustering of these microglia around antigen positive cells, but especially around fragmented cells, suggesting that these cells may play a role in clearance of these cells, and 5) blocking of HSV-1 antigen-specific T cells does not alter the number of reactivating neurons, nor their fate.

This is a highly significant study as it attempts to address the age-old question regarding whether neurons that reactivate HSV latent infection die as a result of the reactivation, or whether they can survive. As the authors point out, there are publications that support both possibilities. In particular, explant-induced reactivation does not tend to show robust evidence for neuronal death as a direct result of reactivation. This is the first study to take this question on looking at reactivation in vivo. The study is very well-designed and carefully performed. Because reactivation is a relatively rare event, large cohorts of mice were examined to allow statistically defendable conclusions to be drawn. This is an outstanding study and one that will be highly cited and have a large impact on the field.

**Part II – Major Issues: Key Experiments Required for Acceptance**

Reviewer #1: (No Response)

Reviewer #3: None. The authors have done a great job of thoughtfully addressing comments raised in the previous review.

**Part III – Minor Issues: Editorial and Data Presentation Modifications**

Reviewer #1: (No Response)

Reviewer #3: None. The authors have done a great job of thoughtfully addressing comments raised in the previous review.

PLOS authors have the option to publish the peer review history of their article (what does this mean?). If published, this will include your full peer review and any attached files.

Reviewer #1: No

Reviewer #3: No

---

## [Editor Report · Acceptance letter]

31 Jan 2020

Dear Dr. Sawtell,

We are delighted to inform you that your manuscript, "Resolution of herpes simplex virus reactivation in vivo results in neuronal destruction," has been formally accepted for publication in PLOS Pathogens.

Best regards,

Kasturi Haldar

Editor-in-Chief

PLOS Pathogens

orcid.org/0000-0001-5065-158X

Michael Malim

Editor-in-Chief

PLOS Pathogens

orcid.org/0000-0002-7699-2064